# GLP-1RA improves diabetic renal injury by alleviating glomerular endothelial cells pyrotosis via RXRα/circ8411/miR-23a-5p/ABCA1 pathway

Weixi Wu[1,2☯], Yao Wang[1,2☯], Xian Shao[1,2☯], Shuai Huang[1,2], Jian Wang[1,2], Saijun Zhou[1,2], Hongyan Liu[1,2], Yao Lin[1,2], Pei Yu[1,2]*

1 NHC Key Laboratory of Hormones and Development, Chu Hsien-I Memorial Hospital and Tianjin Institute of Endocrinology, Tianjin Medical University, Tianjin, China, 2 Tianjin Key Laboratory of Metabolic Diseases, Tianjin Medical University, Tianjin, China

☯ These authors contributed equally to this work.

* yupei@tmu.edu.cn

**Data Availability Statement:** Il relevant data are within the manuscript.

**Funding:** This work was supported by financial support from Tianjin Science and Technology

## Abstract

### Background

Lipotoxicity has been implicated in diabetic kidney disease (DKD). However, the role of high glucose levels in DKD and the underlying renal protective mechanisms of GLP-1 receptor agonists (GLP-1RAs) remain unclear.

### Methods

To investigate cholesterol accumulation, pyroptosis in glomerular endothelial cells (GEnCs), and the renal protective mechanisms of GLP-1RAs, we used various techniques, including RT-qPCR, Oil Red O staining, Western blotting, lactate dehydrogenase (LDH) activity assays, circRNA microarrays, bioinformatics analysis, gain and loss-of-function experiments, rescue experiments, and luciferase assays. Additionally, *in vivo* experiments were conducted using *C57BL/6J* and ApoE-deficient (*ApoE⁻/⁻*) mice.

### Results

GEnCs exposed to high glucose exhibited reduced cholesterol efflux, which was accompanied by downregulation of ATP-binding cassette transporter A1 (ABCA1) expression, cholesterol accumulation, and pyroptosis. Circ8411 was identified as a regulator of ABCA1, inhibiting miR-23a-5p through its binding to the 3'UTR. Additionally, higher glucose levels decreased circ8411 expression by inhibiting RXRα. GLP-1RAs effectively reduced cholesterol accumulation and cell pyroptosis by targeting the RXRα/circ8411/miR-23a-5p/ABCA1 pathway. In diabetic *ApoE⁻/⁻* mice, renal structure and function were impaired, with resulted in increased cholesterol accumulation and pyroptosis; however, GLP-1RAs treatment reversed these detrimental changes.

Major Special Project and Engineering Public Health Science and Technology Major Special Project (No.21ZXGWSY00100), Tianjin Natural Science Foundation Key Project (22JCZDJC00590), Tianjin Key Medical Discipline (Specialty) Construct Project (No.TJYXZDXK-032A), Scientific Research Funding of Tianjin Medical University Chu Hsien-I Memorial Hospital (No.ZXY-ZDSYSZD-1), Whitehorn Diabetes Research Fund Project (No.G-X-2019-56), Technology Project of Sichuan Provincial Health Commission (21PJ127) and Chengdu Medical Research Project (2022291).The funders had no role in study design, data collection and analysis, decision to publish, or preparation of the manuscript.

**Competing interests:** The authors have declared that no competing interests exist.

**Abbreviations:** DKD, diabetic kidney disease; GLP-1RA, GLP-1 receptor agonist; GEnCs, glomerular endothelial cells; IL-1β, interleukin-1β; IL-18, interleukin-18; LDH, lactate dehydrogenase; ABCA1, ATP-binding cassette transporter A1; circRNAs, Circular RNAs; ncRNAs, noncoding RNAs; GLP-1R, Glucagon-like peptide-1 receptor; LG, low glucose; HG, high glucose; ECM, endothelial cell medium; BUN, blood urea nitrogen; Scr, serum creatinine; TC, total cholesterol; LDL-c, low-density lipoprotein cholesterol; HDL-c, high-density lipoprotein cholesterol; TG, plasma triglycerides; UTP, urinary protein quantity; PBS, phosphate buffered saline; TFs, transcription factors.

## Conclusions

These findings suggest that the RXRα/circ8411/miR-23a-5p/ABCA1 pathway mediates the contribution of high glucose to lipotoxic renal injury. Targeting this pathway may represent a potential therapeutic strategy for patients with DKD and hypercholesterolemia. Moreover, GLP-1RAs may provide renal protective effects by activating this pathway.

## Introduction

Diabetic kidney disease (DKD) is a leading cause of end-stage renal disease (ESRD) [1], which poses a significant health and economic burden [2]. Current strategies to prevent or treat DKD have limited efficacy, such as the management of blood glucose, blood pressure, and urinary protein excretion. This limitation can be attributed to an insufficient understanding of the underlying pathophysiological mechanisms [3].

Numerous studies have established a strong association between abnormal lipid metabolism and DKD. The accumulation of triglycerides (TG) and cholesterol in the kidneys has been implicated in the pathogenesis of DKD [4–6]. Lipotoxicity, which is characterized by the dysregulation of intracellular homeostasis due to lipid accumulation, leads to metabolic, inflammatory, and oxidative stress [7]. Hyperlipidemia, oxidized low-density lipoprotein (ox-LDL), and cholesterol crystals have been shown to activate caspase-1 in endothelial cells, thereby triggering pyroptosis [8,9]. Pyroptosis is an inflammatory process that is characterised by the release of pro-inflammatory factors, including interleukin-1β (IL-1β), IL-18, and lactate dehydrogenase (LDH). These factors can contribute to a more pronounced inflammatory response [10]. However, the specific mechanisms underlying pyroptosis in the context of DKD development remain unclear.

There is an increasing consensus that glomerular endothelial cells (GEnCs) play a critical role in the development and progression of DKD [11–13]. The ATP-binding cassette transporter A1 (ABCA1) is a key regulator of cholesterol efflux from cells, promoting the removal of cholesterol [14–16]. Previous researches suggest that ABCA1 expression in GEnCs may help to prevent endothelial dysfunction [5,17].

ABCA1 is a plasma membrane protein that serves as a key regulator of cholesterol efflux from cells[18,19]. *In vitro* studies have demonstrated that stimulation of high glucose can downregulate ABCA1 expression in renal cells, and a similar decrease in ABCA1 has been observed in diabetic and nephrotic mouse models [20,21]. Furthermore, our previous research indicated that the dysregulation of glucose and cholesterol metabolism can impair ABCA1 function [22]. Collectively, these findings suggest that dysregulation of ABCA1 may contribute to cholesterol accumulation in the kidney.

Noncoding RNAs (ncRNAs) are a class of RNAs that do not encode proteins. A specific subset of ncRNAs are known as circular RNAs (circRNAs). CircRNAs are characterized by a covalently closed loop formed through a back-splicing process [23]. CircRNAs can act as microRNA (miRNA) sponges or interact with RNA-binding proteins, thereby regulating gene expression [24]. They have been identified as key regulators in various diseases [25]. However, the potential role of circRNAs in the progression of DKD remains unexplored. The glucose-like peptide-1 receptor (GLP-1R) and its agonists (GLP-1RAs) have shown protective effects against oxidative stress and macrophage activation in DKD [26]. Specifically, Exendin-4, a GLP-1RA, has been shown to ameliorate lipotoxicity-induced injury in GEnCs by enhancing ABCA1-mediated cholesterol efflux [27,28]. Based on these findings, we hypothesize that

GLP-1RAs may exert their renal protective effects in DKD by regulating lipotoxicity-induced pyroptosis in GEnCs through the modulation of ABCA1. Notably, circRNAs may play a significant role in this regulatory process.

## Materials and methods

### Cell culture experiments

The GEnCs (SienCell, United States) were cultured in an endothelial cell culture medium. To investigate the effects of high glucose and high cholesterol on GEnCs, the cultures were divided into four groups: a low glucose control group (LG, 5.5 mmol/L D-glucose), a high glucose group (HG, 25.5 mmol/L D-glucose), a high cholesterol group (HC, 400 μg/ml water-soluble cholesterol, Sigma, United States), and a combined high glucose and high cholesterol group (HG+HC, 25.5 mmol/L D-glucose + 400 μg/ml water-soluble cholesterol).

### Animal experimental design

Animal experiments were conducted with the approval of the Tianjin Medical University Animal Ethics Committee. Male mice of the *C57BL/6J* and *ApoE*[-/-] strains (6 weeks old) were obtained from the Huafukang Animal Center, Beijing, China, and housed at the Laboratory Animal Center of the Tianjin Key Laboratory of Metabolic Diseases, Chu Hsien-I Memorial Hospital. The mice were divided into seven groups: a wild-type control group (WT-NC), a diabetic *C57BL/6J* group(WT-DM), an *ApoE*[-/-] group(*ApoE*[-/-]), a diabetic *ApoE*[-/-] group(*ApoE*[-/-]DM), and three diabetic *ApoE*[-/-] groups treated with liraglutide, loxenatide, or insulin. All groups except the wild-type control and *ApoE*[-/-] groups were fed a Western diet and injected with low-dose streptozotocin (50 mg/kg) in citrate buffer after an overnight fast. Mice with a random blood glucose level exceeding 16.7 mmol/L for three consecutive days were considered diabetic. The wild-type control and *ApoE*[-/-] groups were injected with citrate buffer only. All mice were maintained on their respective diets until the conclusion of the study. The *ApoE*[-/-]DM+Lira group received subcutaneous injections of liraglutide (0.2 nmol/kg/day), while the *ApoE*[-/-]DM+Loxenatide group received subcutaneous injections of loxenatide (0.3 nmol/kg/week) for eight weeks following the induction of diabetes. Moreover, we incorporated an insulin-treated diabetic *ApoE*[-/-] group into our study to investigate whether the beneficial renal effects of GLP-1RAs were independent of their glycemic control capabilities. All mice were subjected to endpoint metabolic cages, where their food consumption, water intake, and urine output were recorded over a 24 hours period. To minimize distress, the mice were closely monitored throughout the study and were euthanized prior to reaching the humane endpoint. The mice were sacrificed by cervical decapitation under anesthesia, utilizing inhalation of 5% isoflurane. Blood and tissue samples were collected and subsequently allocated for various assays. The kidneys were preserved in liquid nitrogen for future analysis.

### Quantification of total cholesterol

Enzymatic colorimetric assays were performed using a cholesterol determination kit from Jiancheng Bioengineering (Nanjing, China) to quantify cholesterol levels in GEnCs. Six-well plates containing GEnCs were subjected to ultrasonic disruption, followed by cell lysis. The resulting lysates were then processed according to the manufacturer's protocol for the total cholesterol assay. Absorbance measurements were taken at a wavelength of 510 nm using a microplate reader to determine the cholesterol concentration in the samples.

## Immunofluorescence

Concentration of 4% Paraformaldehyde was applied to the cells for 30 minutes. This was followed by the application of goat serum at a concentration of 5%. Subsequently, a primary antibody from Abcam, diluted 1:200, was incubated with the cells overnight at a temperature of 4 degrees Celsius. A secondary antibody, a FITC-conjugated goat anti-rat IgG (H+L) obtained from Sun Gene Biotech, China, was then incubated with the cells for a period of 1 hour in a dark environment. The nucleus were stained with 4',6-diamidino-2-phenylindole (DAPI) for 5 minutes. Finally, fluorescence microscopy was employed to visualize the cells.

## Cell death assay

The release of LDH was employed as a biomarker to assess pyroptotic cell death. Cells cultured in six-well plates were subjected to ultrasonic disruption to lyse the cells. Following this, the LDH activity within the lysate was quantified using an LDH assay kit obtained from Jiancheng Bioengineering in Nanjing, China. The manufacturer's guidelines were strictly adhered to during the execution of the assay. To measure the absorbance of the resulting reaction mixture, a microplate reader was utilized, with readings taken at a wavelength of 450 nm.

## Cell transfection

A circ8411-pcDNA3.1 plasmid and a control plasmid, siRNAs targeting circ8411 and its negative control RNA (si-NC), a miR-23a-5p mimic, and its negative control were obtained from GenePharma (Shanghai, China). GEnCs were transfected with these plasmids using the Lipofectamine 2000 system (Thermo Fisher Scientific) according to the manufacturer's instructions. The specific siRNA sequences used are listed in Table 1.

## RNA isolation and quantitative real-time PCR analyses

The Trizol Reagent (Invitrogen, Waltham, the United States) was employed to extract the total RNA from GEnCs following transfection with circ8411-pcDNA3.1 or siRNA, miR-125a-5p mimic, or treatment with drugs. Subsequently, cDNA was synthesized from RNA using a reverse transcription set (Takara, China). Quantitative real-time polymerase chain reactions (RT-qPCR) were conducted using SYBR Green Master (Takara, Beijing, China) and CFX96 real-time PCR detection systems (Bio-Rad, United States), with the primer sequences listed in Table 2.

## Western blot analysis

Protein extraction from treated cells was accomplished by lysing them in a RIPA lysis buffer supplemented with a protease inhibitor cocktail. Subsequently, the proteins were separated on

**Table 1.  The primer sequences for cell transfection.**

| Si-circ8411-1 | 5'-CCACUCUGUACCUGGGACGTT-3' |
|---|---|
|  | 5'-CGUCCCAGGUACAGAGUGGTT-3' |
| Si-circ8411-2 | 5'-GUACCUGGGACGCUGGGGUTT-3' |
|  | 5'-ACGUGACACGUUCGGAGAATT-3' |
| miR-23a-5p mimic | 5'-GGGGUUCCUGGGGAUGGGAUUU-3' |
|  | 5'-AUCCCAUCCCCAGGAACCCCUU-3' |
| miR-23a-5p | 5'-UUCUCCGAACGUGUCACGUTT-3' |
| mimic NC | 5'-ACGUGACACGUUCGGAGAATT-3' |

**Table 2. The primer sequences for RT-qPCR.**

| ABCA1 | 5'-GCAGGCAATCATCAGGGTGC-3' |
|---|---|
|  | 5'-TTCAGCCGTGCCTCCTTCTC-3' |
| Caspase-1 | 5'-TCAGCAGCTCCTCAGGCAGTG-3' |
|  | 5'-AAGACGTGTGCGGCTTGACTTG-3' |
| GSDMD | 5'-GCCAGAAGAAGACGGTCACCATC-3' |
|  | 5'-TTCGCTCGTGGAACGCTTGTG-3' |
| IL-1β | 5'-GCGGCATCCAGCTACGAATCTC-3' |
|  | 5'-AACCAGCATCTTCCTCAGCTTGTC-3' |
| circ1692 | 5'-AGGAGACCATGGTTCCACCC-3' |
|  | 5'-CCTCAACAGGCTGAGCAGGA-3' |
| circ8411 | 5'-AGGAGACCATGGTTCCACCC-3' |
|  | 5'-CCTCAACAGGCTGAGCAGGA-3' |
| circ1477 | 5'-CTGGTCGCACAGGCACAAA-3' |
|  | 5'-TTCGCTCGTGGAACGCTTGTG-3' |
| circ31366 | 5'-GAATAGCTGCGCTTCCCTGG-3' |
|  | 5'-GGAGAGCCACTCAAACATGTTGA-3' |
| RXRα | 5'-AAGATGCGGGACATGCAGAT-3' |
|  | 5'-CGAGAGCCCCTTGGAGTCA-3' |
| HIF-1a | 5'-CTGCCACTGCCACCACAACTG-3' |
|  | 5'-TGCCACTGTATGCTGATGCCTTAG-3' |
| C/EBPα | 5'-TCGGTGGACAAGAACAGCAACG-3' |
|  | 5'-GGCGGTCATTGTCACTGGTCAG-3' |
| GAPDH | 5'-ATGGGGAAGGTGAAGGTCG-3' |
|  | 5'-GGGGTCATTGATGGCAACAATA-3' |

a polyacrylamide gel denatured with sodium dodecyl sulfate (SDS). Following this, the proteins were transferred onto a nitrocellulose (NC) membrane with a concentration of 8–10% SDS. After blocking with 5% skim milk, the membranes were incubated with primary antibodies specific for ABCA1, Caspase-1, GSDMD, N-GSDMD (Abcam), RXRa (ABclonal), and Anti-β-actin (ProteinTech), GAPDH (ABclonal)) at 4°C overnight. Subsequent to washing, the membranes were incubated with secondary antibodies conjugated with horseradish peroxidase (1:3000–5000; Sanjian, Tianjin, China). Finally, Immobilon Western Chemiluminescent HRP Substrate (Thermo Fisher Scientific) was utilized to visualize the protein.

## Microarray analysis

Total RNA was extracted from GEnCs utilizing TRIzol reagent, a commercially available product obtained from Invitrogen, a company headquartered in Waltham, United States. CircRNA microarrays were subsequently analyzed using Affymetrix arrays procured from CapitalBio Corporation, Beijing, China. The microarray hybridization process was conducted in strict adherence to the standardized protocols established by Affymetrix.

## Luciferase assay

The wild-type and mutant circ8411 UTR sequences were inserted into the luciferase reporter vector pmirGLO. Subsequently, these luciferase vectors were transfected into 293T cells expressing either mimic of miR-23a-5p or negative control sequences using Lipofectamine 2000. To assess relative luciferase activity, a Dual Luciferase Reporter Assay kit (Promega, American) was employed. The lysates obtained were analyzed for both firefly and Renilla

luciferase activity. Variations in transfection efficiency were determined by measuring Renilla luciferase activity.

## Biochemical measurements and morphological analysis of kidneys

Blood samples were collected at the study's conclusion through a cardiac puncture procedure using heparinized syringes to prevent clotting. These samples were immediately chilled on ice and then centrifuged at a speed of 5000 g for 10 minutes at a temperature of 4°C. The liquid portion of the blood, known as plasma, was promptly frozen at -80°C for future analysis. To standardize the biochemical measurements obtained from urine samples, the total urine output over a 24-hour period was collected using specialized metabolic cages, as previously described. Blood urea nitrogen (BUN), serum creatinine (Scr), plasma total cholesterol (TC), low-density lipoprotein cholesterol (LDL-c), high-density lipoprotein cholesterol (HDL-c), plasma triglycerides (TG), and urinary protein quantity (UTP) were quantified using a biochemical autoanalyzer equipped with commercial kits. Blood glucose levels were measured with a glucose analyzer.

## Hematoxylin and eosin (H&E) staining

After the study, the kidneys were removed and weighed individually. A portion of the right kidney was preserved in a 4% paraformaldehyde fixation solution for 24 hours, followed by an additional 24 hours in 70% ethanol. After being perfused with 0.01 mol/L PBS for 10 minutes, the tissue was embedded in paraffin and cut into 4 μm sections. Histological examination was conducted using a light microscope, employing H&E staining according to the manufacturer's established protocol.

## Statistical analysis

Data were presented as the "mean ± standard deviation (SD)". To compare two groups, a Student's t-test was employed. For comparisons involving multiple groups, a one-way analysis of variance (ANOVA) was conducted. Statistical significance was defined as a $p$-value less than 0.05. The statistical analysis was performed using Prism 8.0 software (GraphPad, La Jolla, CA, USA).

## Results

### High glucose exacerbates intracellular cholesterol accumulation and pyroptosis under cholesterol load in GEnCs

CCK8 assay and Oil Red O staining were employed to determine the optimal concentration and time for cholesterol. Based on the results, 400 μg/ml cholesterol and 24 hours were identified as optimal intervention concentration and time for GEnCs (Fig 1A and 1B). The treatment of GEnCs with varying concentrations of glucose and cholesterol demonstrated a gradual elevation in ABCA1 mRNA expression with increasing cholesterol levels (Fig 1D), and the reduction in ABCA1 mRNA was most pronounced when cells were treated with 25.5mmo/L glucose (Fig 1E and 1F). Accordingly, 400 μM cholesterol and 25 μM glucose were selected as the intervention concentrations for the subsequent cell experiments. When compared to high glucose or high cholesterol individually, intracellular cholesterol accumulation was significantly elevated in cells treated with a combination of high glucose and high cholesterol (Fig 1G and 1H). Furthermore, the results demonstrated that ABCA1 levels were downregulated in the HG group compared to LG and significantly upregulated in the HC group. ABCA1 expression was significantly downregulated in the HG and HC groups compared to HC (Fig 1I–1K). In

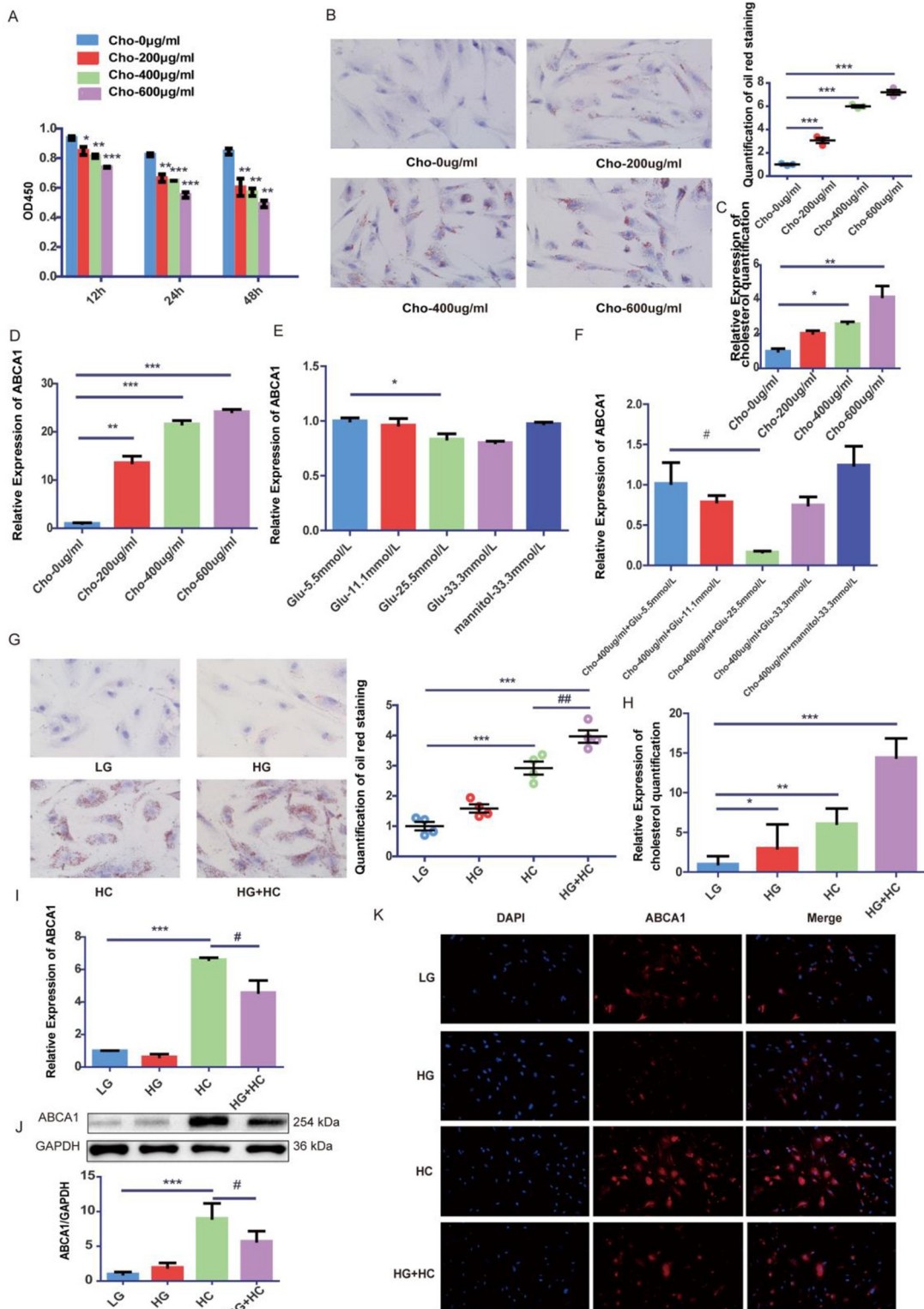

**Fig 1. High glucose exacerbates intracellular cholesterol accumulation under cholesterol load in GEnCs.** (A) CCK-8 assay was used to evaluate the influence of water-soluble cholesterol on GEnCs viability (*P < 0.05 vs. Cho-0μg/ml group; **P < 0.01 vs. Cho-0μg/ml group, ***P < 0.001 vs. Cho-0μg/ml group). (B)Representative images of Oil red O staining of GEnCs treated with different concentrations of cholesterol and the quantitative assessment(×400, bar = 50μm; ***P < 0.001 vs. Cho-0μg/ml group). (C) Cholesterol quantification experiment was conducted to determine the content of intracellular cholesterol

(*P < 0.05 vs. Cho-0μg/ml group; **P < 0.01 vs. Cho-0μg/ml group). (D)The mRNA expression of ABCA1 in GEnCs was detected by RT-qPCR with different cholesterol concentrations (**P < 0.01 vs. Cho-0 μg/ml group, ***P < 0.001 vs. Cho-0μg/ml group). (E) The mRNA expression of ABCA1 in GEnCs was detected by RT-qPCR after intervention with different glucose concentrations without cholesterol(*P < 0.05 vs. Glu-5.5mmol/L group). (F) The mRNA expression of ABCA1 in GEnCs was detected by RT-qPCR after intervention with different glucose concentrations under cholesterol load (#P < 0.05 vs. Cho-400μg/ml+Glu-5.5mmol/L group). (G) Representative images of Oil red O staining of GEnCs and the quantitative assessment(×400, bar = 50μm; ##P < 0.01 vs. HC, ***P < 0.001 vs. LG group). (H) Cholesterol quantification experiment was conducted to determine the content of intracellular cholesterol (*P < 0.05 vs. LG group, **P < 0.01 vs. LG group; ***P < 0.001vs. LG group). (I) RT-qPCR analysis to determine ABCA1 mRNA expression (***P < 0.001 vs. LG group; #P < 0.05 vs. HC group). (J) Western Blot analysis to determine ABCA1 protein expression (***P < 0.001 vs. LG group; #P < 0.05 vs. HC group). (K) Representative images of Immunofluorescence staining of ABCA1(red) in GEnCs(×200, bar = 100μm). Nucleus were stained with DAPI (blue).

summary, these findings suggest that high glucose may exacerbate intracellular cholesterol accumulation under conditions of cholesterol overload, the process may be mediated by ABCA1.

To investigate the influence of cholesterol accumulation on cellular pyroptosis, the occurrence of pyroptosis in GEnCs was examined. RT-qPCR and Western blotting revealed that compared to LG conditions, the expression of pyroptosis markers such as caspase-1, gasdermin D (GSDMD), and IL-1β was elevated in HG or HC conditions. Moreover, a more pronounced increase in pyroptosis markers was observed in the combined HG+HC condition compared to HG or HC alone (Fig 2A and 2B). These findings were further supported by the release of LDH (Fig 2C). To assess the impact of ABCA1 expression on the pyroptosis of GEnCs under cholesterol load, 4,4'-diisothiocyanatostilbene-2,2'-disulfonic acid (DIDS), an inhibitor of ABCA1, was employed to suppress ABCA1 expression. A significantly reduction in ABCA1 expression was observed in cells treated with 400 μmol/L DIDS under high glucose and cholesterol conditions (Fig 2D and 2E). Concurrently, the inhibition of ABCA1 led to cholesterol accumulation (Fig 2F and 2G), accompanied by significant increases in the expression of caspase-1, GSDMD, IL-1β, and the release of LDH (Fig 2H–2J). These results suggest that high glucose may exacerbate pyroptosis in GEnCs under cholesterol load by decreasing the expression of ABCA1.

## Circ8411/miR-23a-5p is involved in the regulation of ABCA1

The heat map presented in Fig 3A illustrates the microarray data of aberrantly expressed circRNAs. The circRNA microarray results revealed that, compared to the HC group, the HG+HC group exhibited 611 downregulated and 279 upregulated circRNAs (|logFC| >1.5) (S1 Table). Among the downregulated circRNAs, four candidate circRNAs (circ1692, circ8411, circ1477, and circ31366) were identified based on a |logFC| threshold of >2.0. To investigate its potential biological role in GEnCs, circ8411 was selected for further study (Fig 3B). To examine whether circ8411 expression is responsive to cellular glucose and cholesterol levels, we evaluated circ8411 expression in GEnCs treated with high glucose and cholesterol. Our findings indicated that circ8411 levels increased in the HC group compared to the LG or HG groups but decreased in the HG+HC group (Fig 3C). Additionally, GEnCs were transfected with small interfering RNAs (siRNAs) targeting circ8411 or a control siRNA, as well as empty vectors and circ8411 plasmids, we observed that circ8411 expression was lower in cells transfected with siRNA-1 or siRNA-2 than the negative control siRNA (NC-si), and siRNA-1 demonstrated superior transfection efficiency (Fig 3D). As depicted in Fig 3J–3L, the knockdown of circ8411 using siRNA-1 under LG or HG+HC conditions significantly decreased ABCA1 expression, exacerbated cholesterol accumulation, increased the expression of caspase-1, GSDMD, and IL-1β, and promoted the release of LDH. Conversely, overexpression of circ8411 in GEnCs yielded the opposite effects (Fig 3M–3T).

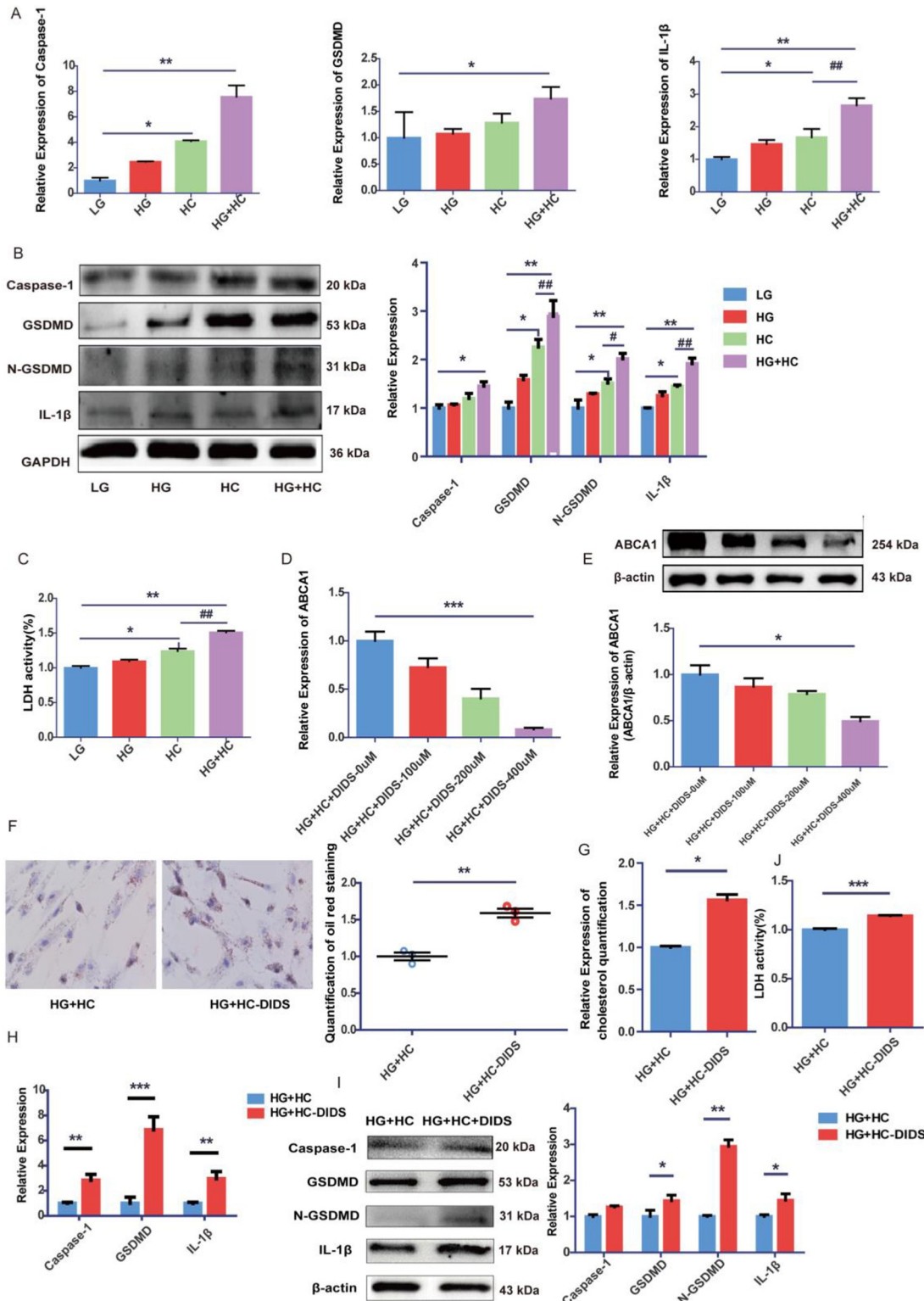

**Fig 2. HG exacerbates GEnCs pyroptosis through ABCA1 under cholesterol load.** (A,B) RT-qPCR and Western Blot analysis to determine expressions of caspase-1, GSDMD, IL-1β in GEnCs under the condition of high glucose or/and high cholesterol (*P < 0.05 vs. LG group,**P < 0.01 vs. LG group; #P < 0.05 vs. HC group, ##P < 0.01 vs. HC group). (C) The release of LDH of GEnCs treated with different conditions(*P < 0.05 vs. LG group, **P < 0.01 vs. LG group; ##P < 0.01 vs. HC group). (D,E) RT-qPCR and Western Blot analysis to determine ABCA1 expression of GEnCs treated with different concentrations of DIDS

($*P < 0.05$ vs. HG+HC+DIDS-0μmol/L group,$***P < 0.001$ vs. HG+HC+DIDS-0μmol/L group). (F) Representative images of Oil red O staining and analysis of GEnCs treated with 400 μmol/L DIDS (×400, bar = 50μm; $**P < 0.01$ vs. HG+HC group). (G) Cholesterol quantification experiment was conducted to determine intracellular cholesterol content ($*P < 0.05$ vs. HG+HC group). (H,I) RT-qPCR and Western Blot analysis to determine expressions of caspase-1, GSDMD, IL-1β in GEnCs treated with 400 μmol/L DIDS ($*P < 0.05$ vs. HG+HC group; $**P < 0.01$ vs. HG+HC group, $***P < 0.001$ vs. HG+HC group). (J) Release of LDH of GEnCs treated with 400 μmol/L DIDS ($***P < 0.001$ vs. HG+HC group).

circRNAs have historically been recognized as miRNA sponges, playing a role in regulating miRNA activity. Consequently, we sought to investigate the potential ceRNA function of circ8411. Yang et al.[29] demonstrated that miR-23a-5p could suppress the expression of ABCA1. In our study, a significant decrease in miR-23a-5p levels was observed in HC, and this change was reversed in HG+HC (Fig 4A). This pattern is in contrast to the expression levels of circ8411 and ABCA1. Bioinformatics analysis suggested that circ8411 might act as a ceRNA for miR-23a-5p. Our findings revealed that knocking down circ8411 led to an increase in miR-23a-5p expression (Fig 4B) while overexpressing circ8411 resulted in a noticeable decrease in miR-23a-5p expression (Fig 4C). Furthermore, miR-23a-5p mimic reduced ABCA1 expression (Fig 4E). We observed that overexpressing miR-23a-5p could repress ABCA1 expression, and this repression was abrogated by the overexpression of circ8411 (Fig 4F and 4G). The results from our luciferase assay showed that the luciferase activity in the circ8411-3'UTR-WT+miR-23a-5p mimic group was significantly lower compared to the circ8411-3'-UTR-WT+miR-23a-5p-NC group. There was no significant difference in luciferase activity between the circ8411-3'UTR-WT+miR-23a-5p mimic and circ8411-3' UTR-WT+miR-23a-5p mimic groups (Fig 4H).

We observed that elevated glucose levels can decrease the expression of circ8411. However, additional research is necessary to fully understand the underlying molecular mechanisms. One potential explanation is that high glucose concentrations may interfere with the transcription of circ8411 by transcription factors (TFs). Bioinformatics tools (Gene, PROMO, JASPAR) predicted and identified three TFs (CEBP/α, RXRα, HIF-1a) as potential regulators. Among these, CEBP/α and RXRα exhibited significantly decreased expression in GEnCs cultured in HG+HC group compared to those cultured in HC group (Fig 5A). RXRα was selected for further investigation in the subsequent study. To suppress RXRα expression effectively in GEnCs, we employed different concentrations of UVI3003. Treatment with 10 μmol/L UVI3003 resulted in a noticeable decrease in RXRα expression (Fig 5B). Moreover, inhibiting RXRα led to downregulation of circ8411α and ABCA1 (Fig 5C–5E). These findings suggested that RXRα may play a role in the transcriptional regulation of circ8411.

## GLP-1RA attenuated cholesterol accumulation and pyroptosis in GEnCs through RXRα-circ8411-miR-23a-5p-ABCA1

We investigated the impact of GLP-1RAs on the expression of ABCA1 in GEnCs exposed to HG+HC. Our findings revealed that liraglutide and loxenatide enhanced the viability of GEnCs in HG+HC and upregulated ABCA1 expression. Specifically, ABCA1 expression was significantly elevated in GEnCs treated with 1000 nmol/L liraglutide and 80 nmol/L loxenatide in HG+HC (Fig 6A–6D). Additionally, we observed that liraglutide and loxenatide treatments decreased cholesterol accumulation and pyroptosis in GEnCs (Fig 6E–6N). Furthermore, we found that the expression of caspase-1, GSDMD, IL-1β, and the release of LDH were reduced in the treatment groups compared to the control group (Fig 6I–6N). These results suggest that liraglutide and loxenatide upregulate ABCA1 expression and ameliorate cholesterol accumulation and pyroptosis in GEnCs. To delve deeper into the underlying mechanisms through which GLP-1RAs influence ABCA1-mediated cholesterol efflux, we examined the expression

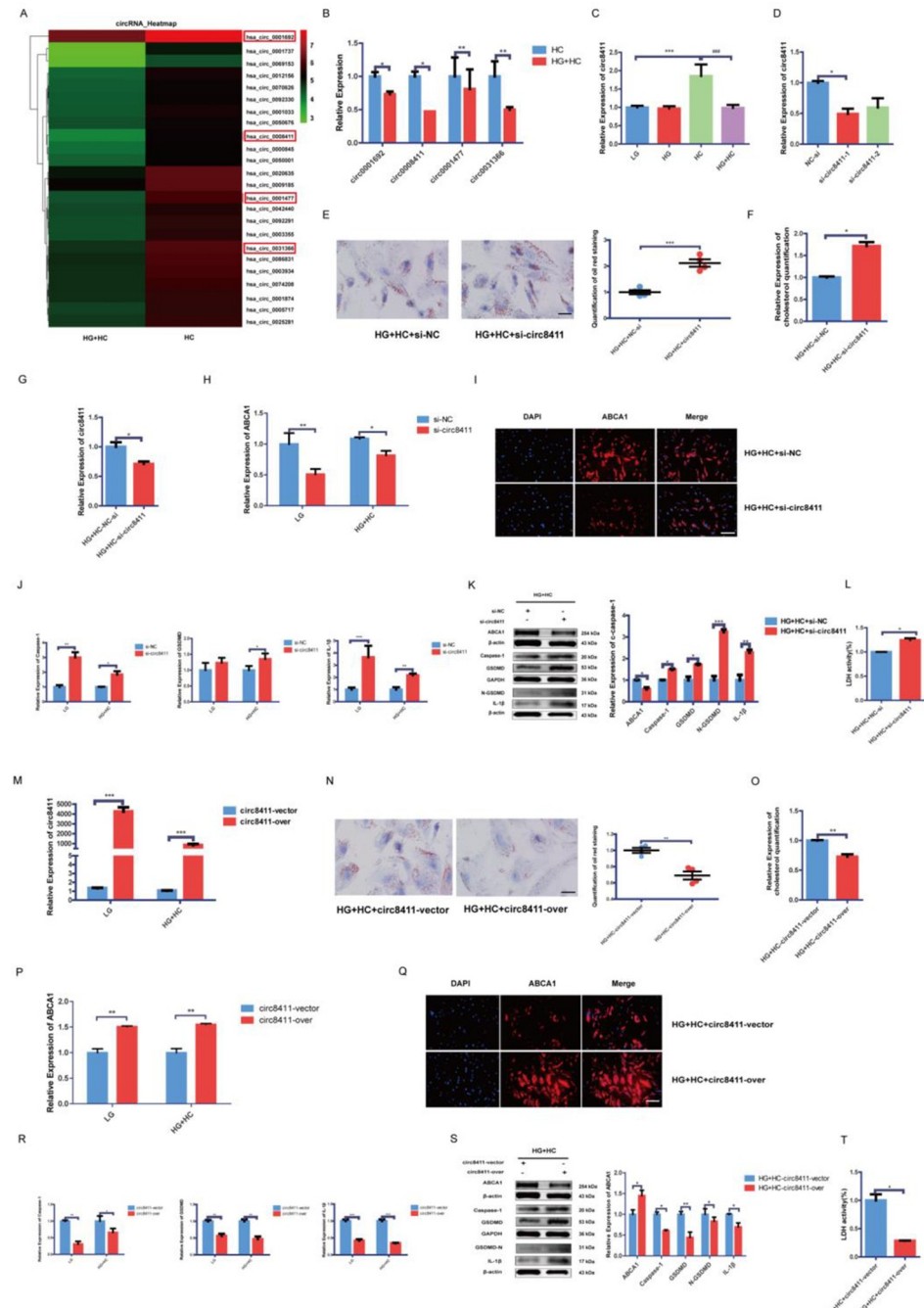

**Fig 3. Circ8411 is involved in the regulation of ABCA1.** (A) The heatmap for the differential expression of circRNAs. (B) RT-qPCR analysis to determine the expression of four candidate lncRNAs (*P < 0.05 vs. HC group; **P < 0.01 vs. HC group). (C) Circ8411 expression of GEnCs treated with cholesterol and glucose (***P < 0.001 vs. LG group, ###P < 0.001 vs. HC group). (D) RT-qPCR to determine the knockdown efficiency of circ8411 (*P < 0.05 vs. NC-si group). (E) Representative images of Oil red O staining(×400, bar = 20μm) and analysis in GEnCs after knockdown circ8411(***P < 0.001 vs. HG+HC+si-NC group). (F) Cholesterol quantification experiment to determine the cholesterol accumulation in GEnCs transfected with siRNA (*P < 0.05 vs. HG+HC+si-NC group). (G-I) RT-qPCR, Western Blot, and immunofluorescence experiment (×200, bar = 50μm) were conducted to determine the expression of ABCA1 in cells transfected with siRNA (*P < 0.05 vs. NC-si group; **P < 0.01 vs. NC-si group). (J) RT-qPCR analysis to determine expressions of caspase-1, GSDMD, IL-1β in GEnCs treated with siRNA (*P < 0.05 vs. NC-si group; **P < 0.01 vs. NC-si group, ***P < 0.001 vs. NC-si group). (K)Representative Western blot images and quantitative data of caspase-1, GSDMD, IL-1β in GEnCs treated with siRNA (*P < 0.05 vs. HG+HC+si-NC group; **P < 0.01 vs. HG+HC+si-NC group, ***P < 0.001 vs. HG+HC+si-NC group). (L) Release of LDH in GEnCs treated with siRNA (*P < 0.05 vs. HG+HC+si-NC group). (M) RT-qPCR analysis to determine the overexpression efficiency

of circ8411 (***P < 0.001 vs. cir8411-vector group). (N) Representative images of Oil red O staining(×400, bar = 20μm) and analysis in GEnCs transfected with circ8411-pcDNA3.1 plasmid(**P < 0.01 vs. HG+HC +cir8411-vector group). (O) Cholesterol quantification experiment to determine the cholesterol accumulation in GEnCs transfected with circ8411-pcDNA3.1 plasmid (**P < 0.01 vs. HG+HC+cir8411-vector group). (P) RT-qPCR were conducted to determine the expression of ABCA1 mRNA in cells transfected with the circ8411-pcDNA3.1 plasmid (**P < 0.01 vs. HG+HC+cir8411-vector group). (Q)Representative images of the immunofluorescence staining of ABCA1(red) in cells transfected with the circ8411-pcDNA3.1 plasmid (×200, bar = 50μm). Nucleus were stained with DAPI (blue). (R) RT-qPCR analysis to determine the expression of caspase-1, GSDMD, and IL-1β in GEnCs treated with circ8411-pcDNA3.1 plasmid (*P < 0.05 vs. cir8411-vector group, **P < 0.01 vs. cir8411-vector group). (S) Representative Western blot images and quantitative data of caspase-1, ABCA1, GSDMD, and IL-1β in GEnCs treated with circ8411-pcDNA3.1 plasmid (*P < 0.05 vs. HG+HC+cir8411-vector group; **P < 0.01 vs. HG +HC+cir8411-vector group). (T) Release of LDH in GEnCs treated with circ8411-pcDNA3.1 plasmid (*P < 0.05 vs. HG+HC+cir8411-vector group).

of RXRα, circ8411, and miR-23a-5p in GEnCs. As depicted in Fig 6O–6T, we demonstrated that liraglutide and loxenatide not only increased the expression of RXRα and circ8411 but also decreased the expression of miR-23a-5p in HG+HC-treated GEnCs. Furthermore, siRNA-mediated knockdown of circ8411 attenuated the upregulatory effect of GLP-1RAs on ABCA1 expression (Fig 6U and 6V).

## Renal injury was attenuated in GLP-1RAs-treated mice

Compared with WT-NC, blood glucose, TC, and LDL-c in the WT-DM, $ApoE^{-/-}$ and $ApoE^{-/-}$DM increased, while HDL-c was decreased. There were a higher TG levels in WT-DM than that in the WT-NC and in $ApoE^{-/-}$DM than that in the $ApoE^{-/-}$. The above trends were reversed partially in the GLP-1RAs group. A notable increase in Scr, BUN and UTP was observed in $ApoE^{-/-}$DM mice, and these changes were partially rescued by liraglutide and loxe-natide treatment in $ApoE^{-/-}$DM mice (Table 3). Renal mesangial hyperplasia and renal tubular vacuolar degeneration were evident in $ApoE^{-/-}$DM compared to $ApoE^{-/-}$. Histological analysis of the kidneys from $ApoE^{-/-}$DM mice revealed that liraglutide and loxenatide improved renal mesangial hyperplasia and renal tubular vacuolar degeneration (Fig 7A).

## GLP-1RAs attenuated cholesterol accumulation and pyroptosis in the kidneys of $ApoE^{-/-}$DM Mice by increasing circ8411-ABCA1 expression

Glomerular cholesterol accumulation was significantly elevated in $ApoE^{-/-}$DM compared to $ApoE^{-/-}$ mice, and this trend was markedly reversed in the GLP-1RA group (Fig 7B and 7C). In $ApoE^{-/-}$DM mice, we observed a significant increase in the levels of caspase-1, GSDMD, IL-1β, and the release of LDH, all of which were partially reversed in the GLP-1RAs group (Fig 7D– 7F). Furthermore, compared to $ApoE^{-/-}$ mice, ApoE$^{-/-}$DM mice exhibited a remarkable increase in the expression of circ8411, and ABCA1, while miR-23a-5p expression was significantly reduced. These changes were reversed in the GLP-1RAs group (Fig 7E–7L). These results suggest that hyperglycemia exacerbates renal cholesterol accumulation and pyroptosis in $ApoE^{-/-}$ mice, and these effects can be mitigated by GLP-1RAs through the regulation of the RXRα-circ8411-miR-23a-5p-ABCA1 pathway. Finally, we examined ABCA1 expression in the insulin (INS) group and found no significant increase in ABCA1 expression (S1 Fig). These findings suggest that controlling hyperglycemia alone may not be sufficient to increase ABCA1 expression in diabetic kidneys.

## Discussion

It has been reported that glucotoxicity and lipotoxicity synergistically contribute to renal injury in DKD [29]. This study investigated the mechanisms underlying the effects of high

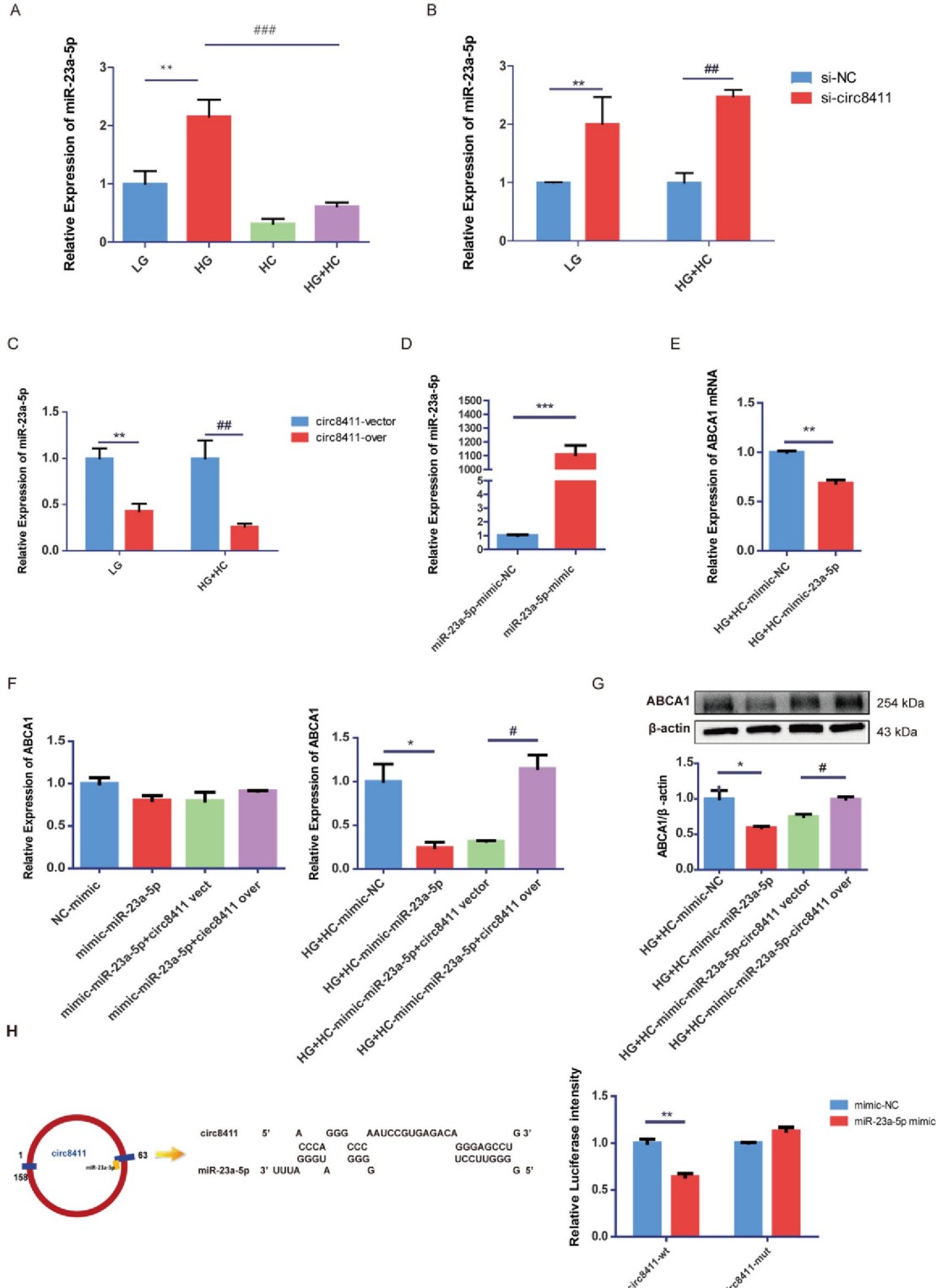

**Fig 4. The circ8411 regulates ABCA1 expression by regulating miR-23a-5p.** RT-qPCR analysis to determine miR-23a-5p in the GEnCs under different conditions (**P < 0.01 vs. LG group; ###P < 0.001 vs. HG group). (B) RT-qPCR analysis to determine miR-23a-5p expression with the knockdown of circ8411(**P < 0.01 vs. LG+si-NC group, ##P < 0.01 vs. HG+HC+si-NC group). (C) RT-qPCR analysis to determine miR-23a-5p expression with the overexpression of circ8411 (**P < 0.01 vs. LG +circ8411-vector group, ##P < 0.01 vs. HG+HC+circ8411-vector group). (D) RT-qPCR analysis to determine the overexpression

efficiency of miR-23a-5p in cells treated with the miR-23a-5p mimic (***P < 0.001 vs. miR-23a-5p mimic-NC group). (E) RT-qPCR analysis to determine the expression of ABCA1 in cells treated with the miR-23a-5p mimic (**P < 0.01 vs. HG+HC+mimic-NC group). (F) Rescue experiments to determine the regulatory effect of circ8411/miR-23a-5p/ABCA1 in GEnCs (*P < 0.05 vs. HG+HC+mimic-NC group, #P < 0.05 vs. HG+HC+mimic-miR-23a-5p+circ8411 vector group). (G) Western blot analysis to determine expression of ABCA1 in rescue experiments(*P < 0.05 vs. HG+HC+mimic-NC group; #P < 0.05 vs. HG+HC+mimic-miR-23a-5p+circ8411 vector group). (H) Luciferase reporter assay on GEnCs co-transfected with circ8411-3'UTR-WT or circ8411-3'UTR-MUT and miR-23a-5p mimic or NC (**P < 0.01 vs. mimic-NC group).

glucose on lipotoxic renal injury. We discovered that high glucose exacerbated cholesterol accumulation and pyroptosis in GEnCs exposed to cholesterol load, as well as in the kidneys of ApoE-deficient diabetic mice, by suppressing ABCA1 expression. We further elucidated that

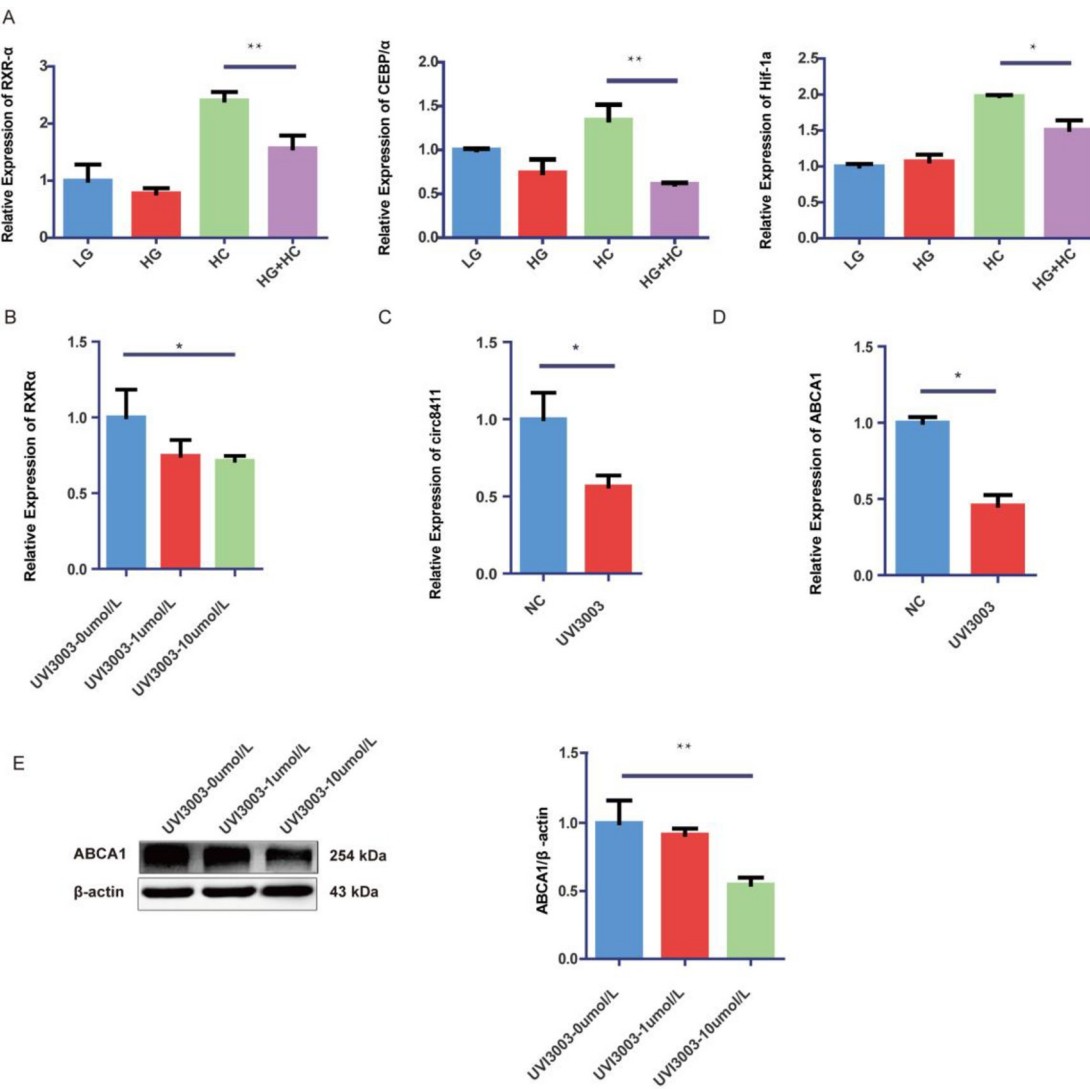

**Fig 5. High glucose affects circ8411 down-regulation through RXRα.** (A) RT-qPCR analysis to determine the expression of RXRα, CEBP/α, Hif-1a in the GEnCs undering glucose and cholesterol (*P < 0.05 vs. HC group, **P < 0.01 vs. HC group). (B) The mRNA expression of RXRα in GEnCs treated with different concentrations of UVI3003 (*P < 0.05 vs. UVI3003-0μmol/L group). (C) The mRNA expression of Circ8411 in GEnCs treated with 10 μmol/L UVI3003(**P < 0.01 vs. NC group). (D)The mRNA expression of ABCA1 expression in GEnCs treated with 10 μmol/L UVI3003 (*P < 0.05 vs. NC group). (E) Representative Western blot images and quantitative data of ABCA1 in GEnCs treated with 10 μmol/L UVI3003(**P < 0.01 vs. UVI3003-0μmol/L group).

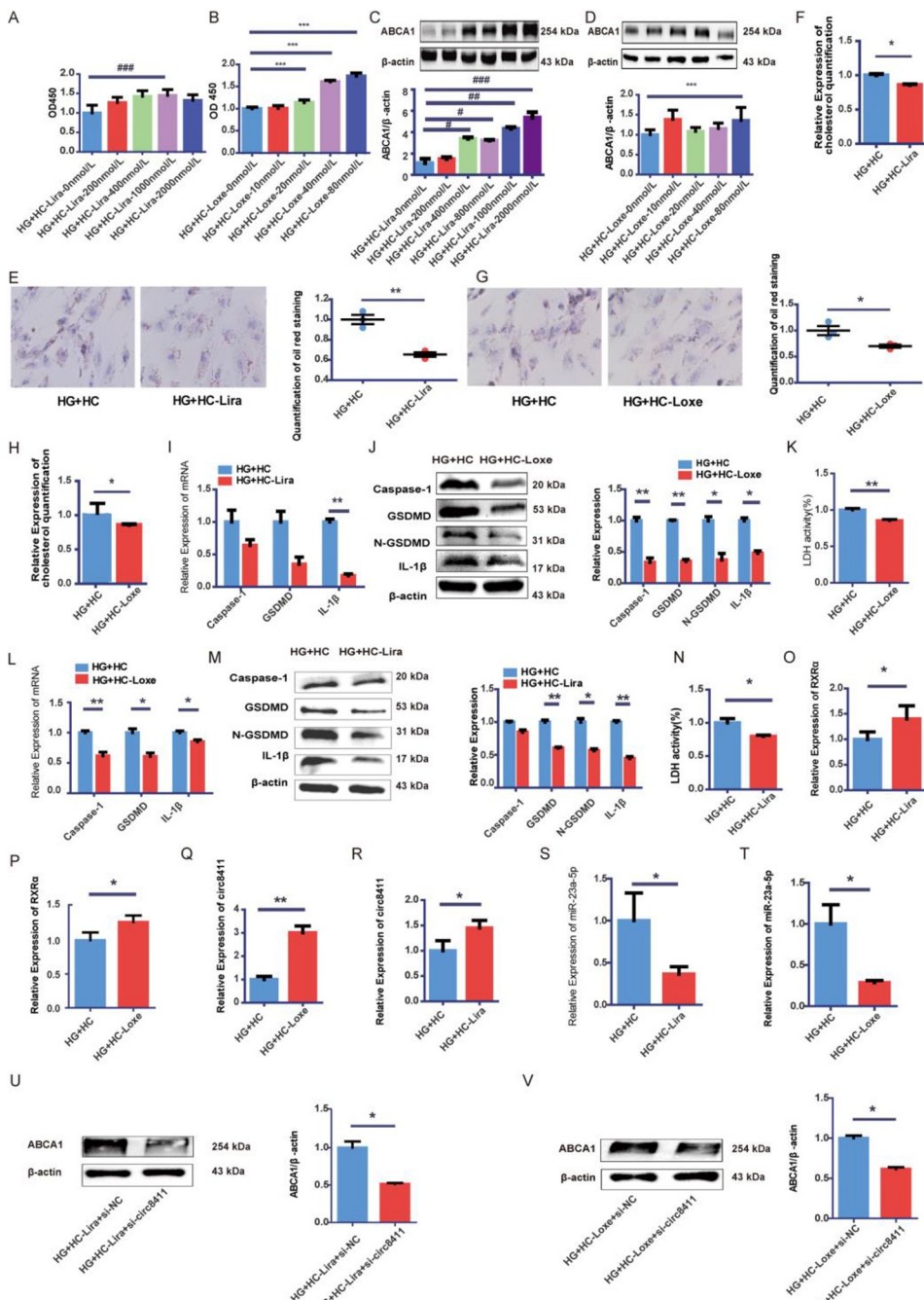

**Fig 6. GLP-1RAs attenuated cholesterol accumulation and pyroptosis in GEnCs by increasing RXRα-circ8411-miR-23a-5p-ABCA1.** (A, B) CCK-8 assay was used to evaluate the influence of liraglutide and loxenatide on GEnC viability (\*\*\*P < 0.001 vs. HG+HC+Loxe-0nmol/L group, ###P < 0.001 vs. HG+HC+Lira-0nmol/L group). (C) ABCA1 expression in the GEnCs treated with different concentrations of liraglutide (#P < 0.05 vs. HG+HC+Lira-0nmol/L group, ##P < 0.01 vs. HG+HC+Lira-0nmol/L group, ###P < 0.001 vs. HG+HC+Lira-0nmol/L group). (D) ABCA1 expression in the GEnCs treated with different concentrations of loxenatide (\*\*\*P < 0.001 vs. HG+HC+Loxe-0nmol/L group). (E, G) Representative images of Oil red O staining(×400, bar = 20μm) and quantitative analysis in GEnCs treated with liraglutide and loxenatide(\*P < 0.05 vs. HG+HC group, \*\*P < 0.01 vs. HG+HC group). (F,H) Cholesterol quantification experiment to determine the cholesterol accumulation in cells treated with liraglutide and loxenatide (\*P < 0.05 vs. HG+HC group). (I) The mRNA expression of caspase-1, GSDMD, IL-1β in GEnCs treated with liraglutide (\*\*P < 0.01 vs. HG+HC group). (J) Representative Western blot images and quantitative data of caspase-1, GSDMD, N-GSDMD, IL-1β in GEnCs treated with loxenatide (\*P < 0.05 vs. HG+HC group, \*\*P < 0.01 vs. HG+HC group). (L) The mRNA expression of caspase-1, GSDMD, IL-1β in GEnCs treated with loxenatide (\*P < 0.05

vs. HG+HC group, **P < 0.01 vs. HG+HC group). (M) Representative Western blot images and quantitative data of caspase-1, GSDMD, N-GSDMD, IL-1β in GEnCs treated with liraglutide (*P < 0.05 vs. HG+HC group, **P < 0.01 vs. HG+HC group). (K,N) Release of LDH of GEnCs treated with loxenatide and liraglutide (*P < 0.05 vs. HG+HC group, **P < 0.01 vs. HG+HC group). (O-T) Expressions of RXRα, circ8411, and miR-23a-5p of GEnCs treated with liraglutide and loxenatide (*P < 0.05 vs. HG+HC group, **P < 0.01 vs. HG+HC group). (U) Representative images of Western blot and quantitative data of the ABCA1 expression in GEnCs treated with si-RNA of circ8411 and liraglutide (*P < 0.05 vs. HG+HC+Lira+si-NC group).(V) Representative images of Western blot and quantitative data of the ABCA1 expression in GEnCs treated with si-RNA of circ8411 and loxenatide (*P < 0.05 vs. HG+HC+Loxe+si-NC group).

high glucose downregulates RXRα and circ8411, thereby mediating the suppression of ABCA1 expression. Knockdown of circ8411 also downregulated ABCA1 expression, leading to aggravated cholesterol accumulation and pyroptosis (Fig 8). Luciferase reporter assays and rescue experiments revealed that circ8411 exerts these effects by binding to miR-23a-5p. Emerging evidence suggests that GLP-1RAs possess extra-renal protective effects [26]. However, the precise mechanisms underlying these renal protective effects remain unclear. Experimental studies have demonstrated that GLP-1RAs increase ABCA1 expression, improve ABCA1-mediated cholesterol efflux, and mitigate pyroptosis (Fig 8). In alignment with these findings, we observed that GLP-1RAs significantly ameliorated renal structural and functional damage, cholesterol accumulation, and pyroptosis in ApoE-deficient diabetic mice.

Cholesterol is now understood to be primarily utilized by the liver and certain glands, while other cells are unable to directly utilize cholesterol and can only maintain cholesterol homeostasis by excreting excess cholesterol. When the cholesterol load exceeds the limits of excretion, cellular dysfunction occurs [5]. Our study revealed that high glucose induced cholesterol accumulation in GEnCs, impairing their ability to respond to cholesterol load. Subsequently, we investigated the mechanisms underlying high glucose-induced intracellular cholesterol accumulation under cholesterol load. Previous studies have demonstrated that ABCA1 expression safeguards against endothelial dysfunction[17]. In the kidney, increased cholesterol and fatty acid synthesis lead to enhanced cholesterol efflux through ABCA1 [19,20]. However, the expression level of ABCA1 in GEnCs and its role in GEnC injury remained unclear. Our findings indicated that ABCA1 expression was upregulated in GEnCs subjected to high cholesterol levels, likely reflecting a cellular response to increased cholesterol content aimed at promoting the efflux of intracellular cholesterol. However, high glucose inhibited the ABCA1 feedback

**Table 3. Biochemical characteristics of plasma and urine of mice.**

| Values | WT-NC | WT-DM | ApoE-/- | ApoE-/-DM | ApoE-/-DM+Lira | ApoE-/-DM+Loxe |
|---|---|---|---|---|---|---|
| Weight (g) | 28.20±1.20 | 27.21±0.14 | 30.77±0.90[a] | 25.84±2.02[b] | 25.12±2.13 | 22.4±2.86[c] |
| FBG (mmol/L) | 5.54±0.50 | 21.20±2.97[a] | 5.86±0.35 | 19.75±2.74[b] | 10.58±3.50[c] | 9.12±4.51[c] |
| TG (mmol/L) | 1.4±0.30 | 2.6±0.50[a] | 1.02±0.30 | 3.15±00.56[b] | 1.40±0.66[c] | 1.02±0.07[c] |
| TC (mmol/L) | 1.57±0.10 | 3.10±1.08[a] | 21.63±2.53[a] | 49.63±6.04[b] | 38.97±5.62 [c] | 41.17±2.91[c] |
| LDL-c (mmol/L) | 1.35±0.20 | 5.73±0.37[a] | 8.10±0.08[a] | 10.37±1.48[b] | 8.34±0.37[c] | 9.43±0.01 |
| HDL-c (mmol/L) | 0.95±0.17 | 1.43±0.06[a] | 1.79±0.02[a] | 1.53±0.40[b] | 1.87±0.51[c] | 2.2±0.18[c] |
| BUN (mmol/L) | 6.02±2.01 | 12.77±1.28[a] | 8.44±0.59[a] | 15.64±0.28[b] | 6.35±2.45[c] | 6.16±0.20[3c] |
| Scr (umol/L) | 12.50±0.40 | 21.30±2.39[a] | 29.75±1.48[a] | 39.9±5.09[b] | 27.65±2.05[c] | 35.5±6.92[c] |
| UTP (mg/d) | 0.44±0.19 | 1.54±0.18[a] | 0.89±0.10[a] | 2.28±0.53[b] | 1.63±0.20[c] | 0.96±0.15[c] |

Values represent the mean± SD

[a]<0.05 vs. WT-NC

[b]<0.05 vs. ApoE-/-

[c]<0.05 vs. ApoE-/-DM.

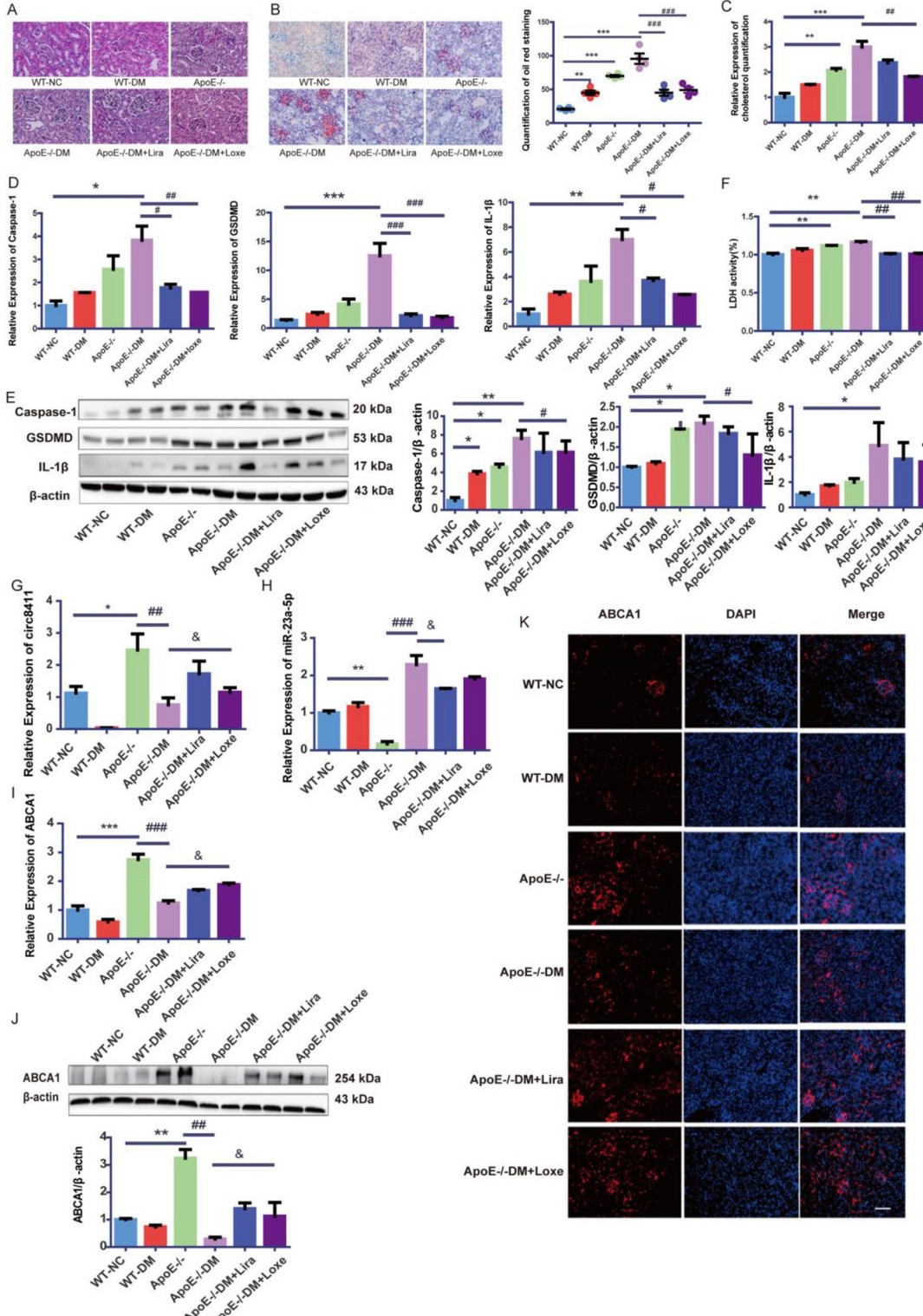

**Fig 7. GLP-1RAs alleviated kidney damage, cholesterol accumulation, and pyroptosis in the kidneys of *ApoE*^-/- DM mice by increasing circ8411-ABCA1.** (A) Representative images of HE staining of the kidney tissues of mice (×400, bar = 50μm). (B) Representative images of Oil red O staining and the quantitative assessment of the kidney tissues of mice (×400, bar = 50μm, **P < 0.01 vs. WT-NC group, ***P < 0.001 vs. WT-NC group, ###P < 0.001 vs. *ApoE*^-/- DM group). (C) Cholesterol quantification experiment to determine the cholesterol accumulation in the kidneys of mice (**P < 0.01 vs. WT-NC group,

***P < 0.001 vs. WT-NC group, ##P < 0.01 vs. *ApoE*[-/-]DM group).(D) The mRNA expression of caspase-1, GSDMD, IL-1β in the kidneys of mice (*P < 0.05 vs. WT-NC group, **P < 0.01 vs. WT-NC group,***P < 0.001 vs. WT-NC group, #P < 0.05 vs. *ApoE*[-/-]DM group, ##P < 0.01 vs. *ApoE*[-/-]DM group, ###P < 0.001 vs. *ApoE*[-/-]DM group). (E) Representative Western blot images and quantitative data of of caspase-1, GSDMD, IL-1β in the kidneys of mice (*P < 0.05 vs. WT-NC group, **P < 0.01 vs. WT-NC group, #P < 0.05 vs. *ApoE*[-/-]DM group).(F) Release of LDH in the kidneys of mice (**P < 0.01 vs. WT-NC group, ##P < 0.01 vs. *ApoE*[-/-]DM group). (G) Circ8411 expression of renal of mice (*P < 0.05 vs. WT-NC group, ##P < 0.01 vs. *ApoE*[-/-] group, &P < 0.05 vs. *ApoE*[-/-]DM group). (H) miR-23a-5p expression of renal of mice (**P < 0.01 vs. WT-NC group, ###P < 0.001 vs. *ApoE*[-/-] group, &P < 0.05 vs. *ApoE*[-/-]DM group). (I-J) RT-qPCR and Western Blot experiment were conducted to determine ABCA1 expression in the kidneys of mice (**P < 0.01 vs. WT-NC group, ***P < 0.001 vs. WT-NC group, ##P < 0.01 vs. *ApoE*[-/-] group, ###P < 0.001 vs. *ApoE*[-/-] group,&P < 0.05 vs. *ApoE*[-/-]DM group). (K) Representative images of the immunofluorescence staining(×200, bar = 50μm) of ABCA1(red) in the kidneys of mice. nucleus were stained with DAPI (blue).

response triggered by high cholesterol, resulting in cholesterol accumulation in GEnCs. These results aligned with our *in vivo* experiments. Collectively, these findings confirm that defects in ABCA1 contribute to cholesterol accumulation in GEnCs.

Excessive cholesterol accumulation can impair cellular functions by inducing inflammation, oxidative stress, and apoptosis. This can lead to damage of renal structure and function. Numerous mechanisms link cholesterol accumulation to renal injury in DKD, with pyroptosis being one of them. Inflammasome-activated caspase-1 plays a pivotal role in inducing pyroptosis. Additionally, a series of studies have confirmed that cholesterol can activate caspase-1 in vascular endothelial cells, triggering pyroptosis [8,30]. However, the relationship between pyroptosis and DKD in GEnCs and the underlying mechanisms remain to be fully elucidated. Our results demonstrated that high glucose upregulated the expression of caspase-1, GSDMD, IL-1β, and LDH release in GEnCs treated with high cholesterol. These levels were also elevated in *ApoE*[-/-]DM mice compared to *ApoE*[-/-]mice. These findings suggest that high glucose can exacerbate intracellular pyroptosis in GEnCs under conditions of excess cholesterol.

Emerging evidence suggests a link between circRNAs and cholesterol metabolism, as well as DKD. For instance, circ0092317, circ0003546, and circACC1 may reduce cholesterol efflux from macrophages [31]. Additionally, CircRNA15698 exacerbates diabetic nephropathy by upregulating extracellular matrix production in mesangial cells through the miR-185/TGF-β pathway [32]. Through circRNA microarray analysis, our study identified a novel circRNA, circ8411, involved in regulating cholesterol homeostasis. We observed a decrease in circ8411 levels in the renal tissues of *ApoE*[-/-]DM mice compared to *ApoE*[-/-]mice. Furthermore, high cholesterol levels could upregulate circ8411 expression, which was partially reversed by high glucose, suggesting that its expression is regulated in response to cellular cholesterol changes. Our study hypothesized a role for circ8411 in the response of GEnCs to cholesterol overload, mediated by the upregulation of cholesterol efflux.

To investigate this, we performed loss-of-function and gain-of-function experiments using siRNA and plasmid transfection. Knockdown of circ8411 significantly suppressed the expression of ABCA1, aggravated cholesterol accumulation, and increased the expression of caspase-1, GSDMD, IL-1β, and LDH release in GEnCs treated with high glucose and high cholesterol. Conversely, overexpression of circ8411 led to the opposite effects. CircRNAs can regulate gene expression by competing for miRNA or protein binding. Based on our findings, we propose that circ8411 acts as an endogenous sponge of miRNA, downregulating miRNA expression and thereby removing the suppression of its target gene ABCA1. In the present study, bioinformatics analysis suggested that miR-23a-5p might serve as a mediator between circ8411 and ABCA1. Dual-luciferase reporter assays demonstrated that miR-23a-5p could repress the activity of the 3' untranslated region (UTR) of circ8411. Additionally, Yang et al. reported that miR-23a-5p binds to ABCA1 [33]. Furthermore, we observed that overexpression of circ8411 downregulated miR-23a-5p. Rescue experiments revealed that overexpression of miR-23a-5p

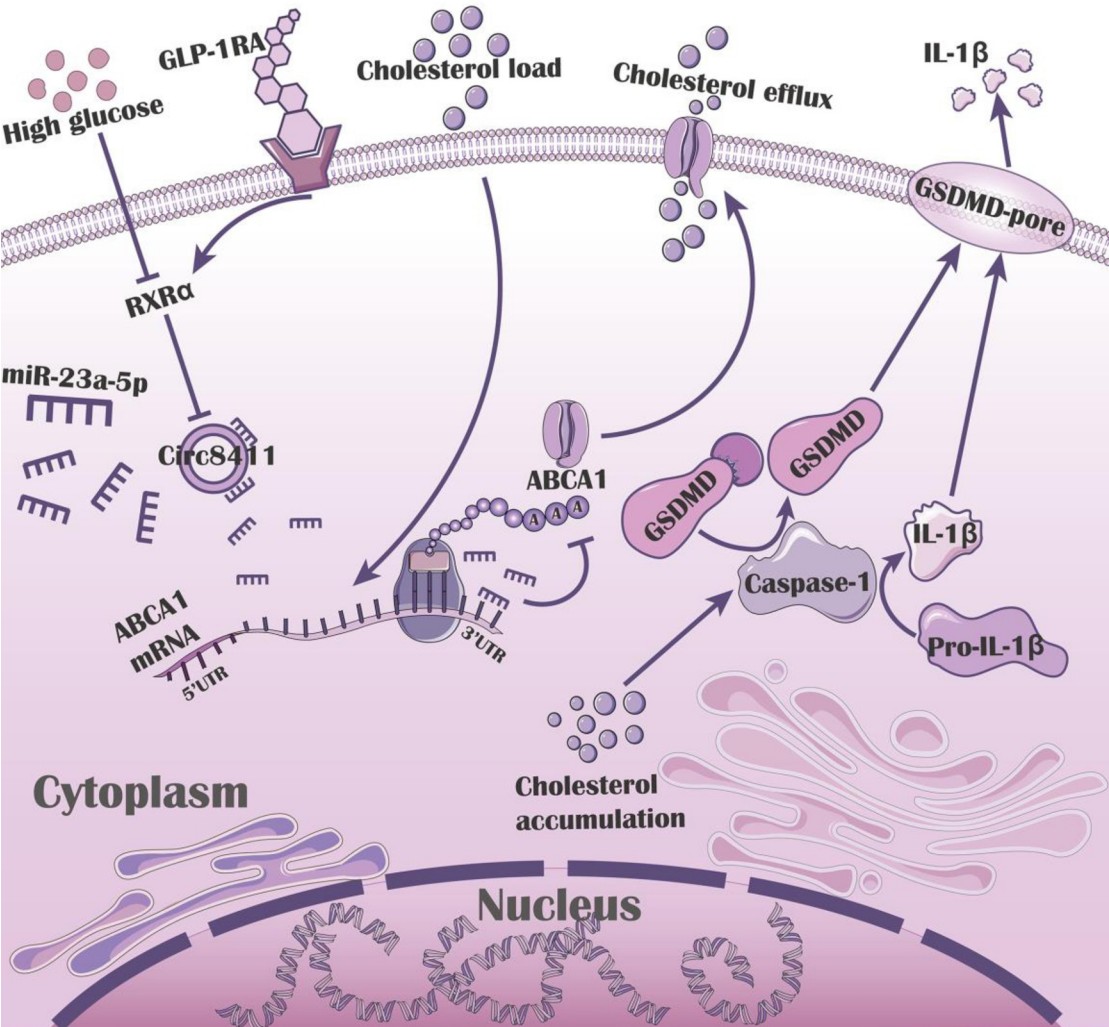

**Fig 8. Proposed model for the signaling pathway by which high glucose participates in the GEnCs injury via the RXRα/ circ8411/miR-23a-5p/ABCA1 pathway.** High glucose decreased the ability of GEnCs to respond to cholesterol load, leading to downregulation of ABCA1 expression, increased cholesterol accumulation, and pyroptosis in GEnCs. Circ8411 regulated ABCA1 by inhibiting the action of miR-23a-5p, and miR-23a-5p binds to the 3' UTR of circ8411. High glucose could decrease circ8411 expression by inhibiting RXRα. GLP-1RAs could potentially reduce cholesterol accumulation and cell pyroptosis by modulating the RXRα/circ8411/miR-23a-5p/ABCA1 axis.

decreased ABCA1 expression, which was partially increased by the addition of circ8411 over-expression, suggesting that circ8411 positively upregulated the expression of ABCA1 in GEnCs by sponging miR-23a-5p. Research on circRNAs primarily focuses on miRNA sponging, transcription, and translation. However, few circRNAs have been utilized as target molecules to investigate their upstream transcriptional regulation. Existing studies have shown that TFs are involved in the transcriptional regulation of circRNAs, such as the TF Twist1, which promotes CUL2 circRNA transcription [34]. To explore the mechanisms underlying high glucose regulation of circ8411, bioinformatics analysis was conducted to identify RXRα as a potential TF of circ8411.

Moreover, RXRα plays a crucial role in phosphorus homeostasis, basal metabolism, and cholesterol metabolism [35]. In our study, we found that high glucose could decrease RXRα expression under cholesterol overload. Inhibition of RXRα using UVI3003 significantly suppressed the expression levels of circ8411 and ABCA1. These findings suggest that RXRα may be involved in the regulation of circ8411 by high glucose. Taken together, we propose that high glucose suppresses the expression of circ8411 by decreasing RXRα expression, ultimately promoting cholesterol accumulation and pyroptosis through the upregulation of miR-23a-5p and subsequent inhibition of ABCA1 expression.

GLP-1R activation stimulates the cAMP and PKA pathways, exerting antioxidant functions, suggesting that GLP-1RAs protects various tissues from oxidative stress [36]. Despite these findings, it remains unclear whether the progression of renal injury under conditions of chronic hyperglycemia is influenced by alterations in GLP-1R signaling. Therefore, we investigated the protective effects of GLP-1RAs on the kidney from the perspective of glucose-lipid metabolism interactions. In our study, we observed that liraglutide and loxenatide exhibited renal protective and lipid-regulating effects in $ApoE^{-/-}$DM mice. These GLP-1RAs decreased cholesterol accumulation in $ApoE^{-/-}$DM mice and GEnCs treated with high glucose and high cholesterol. Additionally, liraglutide and loxenatide inhibited the expression of caspase-1, GSDMD, IL-1β, and LDH release. These findings suggest that GLP-1RAs might mitigate the lipotoxic renal effects aggravated by high glucose. Some studies have indicated that exendin-4 can activate GLP-1R in THP-1 macrophages, thereby increasing ABCA1 expression and preventing the formation of atherogenic foam cells [37,38].

Our study demonstrated that liraglutide and loxenatide could increase ABCA1 expression, and the inhibition of ABCA1 led to increased cholesterol accumulation and pyroptosis in GLP-1RAs-treated GEnCs, indicating that ABCA1 plays a crucial role in the anti-lipotoxicity effects of liraglutide and loxenatide. Yin et al. demonstrated that exendin-4 regulated ABCA1 through GLP-1R-mediated signaling [14]. We provided compelling evidence that liraglutide and loxenatide regulate ABCA1 expression through the RXRα/circ8411/miR-23a-5p pathway. Our findings revealed that GLP-1RAs treatment increased the expression of RXRα and circ8411 while decreasing miR-23a-5p. These effects were inhibited by UVI3003 and siRNA, respectively. Collectively, our results suggest that GLP-1RAs can upregulate ABCA1 expression by modulating the RXRα/circ8411/miR-23a-5p axis, thereby reducing cholesterol accumulation and ameliorating pyroptosis.

In summary, we have presented hitherto undocumented evidence that high glucose exacerbates cholesterol accumulation and pyroptosis in GEnCs, contributing to lipotoxic renal injury. Our study revealed that high glucose significantly downregulates ABCA1 expression in GEnCs and the kidneys of $ApoE^{-/-}$DM mice. Importantly, we have identified that liraglutide and loxenatide exert renal protective effects by reducing cholesterol accumulation and pyroptosis, at least partially through the RXRα/circ8411/miR-23a-5p/ABCA1 pathway under diabetic conditions. Our findings underscore the critical role of lipotoxicity in GEnCs in the development and progression of DKD. Therapeutic agents that enhance GLP-1R activity may offer protection against GEnC dysfunction and the progression of DKD through this pathway.

## Supporting information

**S1 Fig. ABCA1 expression in the INS group.**
(DOCX)

**S1 Table. The results of circRNA microarray.**
(XLSX)

**S1 Raw images.**
(PDF)

## Acknowledgments

We would like to express our deepest gratitude to the animals participated in the research.

## Author Contributions

**Conceptualization:** Weixi Wu, Yao Wang, Saijun Zhou, Pei Yu.

**Data curation:** Weixi Wu, Yao Wang, Xian Shao, Shuai Huang, Yao Lin.

**Funding acquisition:** Pei Yu.

**Investigation:** Weixi Wu, Yao Wang, Xian Shao, Shuai Huang, Jian Wang.

**Methodology:** Weixi Wu, Yao Wang, Pei Yu.

**Project administration:** Pei Yu.

**Resources:** Pei Yu.

**Supervision:** Hongyan Liu, Pei Yu.

**Validation:** Weixi Wu, Yao Wang, Xian Shao, Shuai Huang.

**Visualization:** Weixi Wu, Yao Wang, Xian Shao.

**Writing – original draft:** Weixi Wu, Yao Wang.

**Writing – review & editing:** Xian Shao.

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
