## [Decision Letter · Decision Letter 0]

4 Jun 2024

PONE-D-24-13952GLP-1 RA improves diabetic renal injury by alleviating Glomerular Endothelial Cells Pyrotosis via RXRα/circ8411/miR-23a-5p/ABCA1 pathway PLOS ONE

Dear Dr. Yu,

Thank you for submitting your manuscript to PLOS ONE. After careful consideration, we feel that it has merit but does not fully meet PLOS ONE’s publication criteria as it currently stands. Therefore, we invite you to submit a revised version of the manuscript that addresses the points raised during the review process.

We look forward to receiving your revised manuscript.

Kind regards,

Ravikanth Nanduri, Ph. D.

Academic Editor

PLOS ONE

2. o comply with PLOS ONE submissions requirements, in your Methods section, please provide additional information regarding the experiments involving animals and ensure you have included details on (1) methods of sacrifice, (2) methods of anesthesia and/or analgesia, and (3) efforts to alleviate suffering.

“This work was supported by financial support from Tianjin Science and Technology Major Special Project and Engineering Public Health Science and Technology Major Special Project (No.21ZXGWSY00100), Tianjin Natural Science Foundation Key Project (22JCZDJC00590), Tianjin Key Medical Discipline (Specialty) Construct Project (No.TJYXZDXK-032A), Scientific Research Funding of Tianjin Medical University Chu Hsien-I Memorial Hospital (No.ZXY-ZDSYSZD-1), Whitehorn Diabetes Research Fund Project (No.G-X-2019-56), Technology Project of Sichuan Provincial Health Commission (21PJ127) and Chengdu Medical Research Project (2022291).”

7. We notice that your supplementary tables are uploaded with the file type 'Figure'. Please amend the file type to 'Supporting Information'. Please ensure that each Supporting Information file has a legend listed in the manuscript after the references list.

Additional Editor Comments:

This manuscript need serious English language corrections. Most of the manuscript is not clear. Abbreviations need to be expanded. Authors are requested to answer all the comments of the reviewer.

Profession language proof reading is essential.

Reviewers' comments:

Reviewer's Responses to Questions

**Comments to the Author**

1. Is the manuscript technically sound, and do the data support the conclusions?

Reviewer #1: Yes

Reviewer #2: Partly

2. Has the statistical analysis been performed appropriately and rigorously? 

Reviewer #1: Yes

Reviewer #2: Yes

3. Have the authors made all data underlying the findings in their manuscript fully available?

Reviewer #1: Yes

Reviewer #2: Yes

4. Is the manuscript presented in an intelligible fashion and written in standard English?

Reviewer #1: Yes

Reviewer #2: No

5. Review Comments to the Author

Reviewer #1: These paper about “GLP-1 RA improves diabetic renal injury by alleviating Glomerular Endothelial Cells Pyrotosis via RXRα/circ8411/miR-23a-5p/ABCA1 pathway “suggesting that excessive hyperglycemia causes lipotoxic renal damage through the RXRα/circ8411/miR-23a-5p/ABCA1 43 pathway. Individuals suffering from hypercholesterolemia and diabetic kidney disease (DKD) may find relief by focusing on this pathway.

The paper is well planned and written to put all the relevant data together. There are few minor corrections to be done.

1. The font size throughout the paper is not the same.

2. The spacings in the paper should be modified to be the same throughout the paper.

3. It would be better if you can reduce the thickness of the error bars in all the figures.

4. In Fig 7K immunostaining panel please mention blue and red in the figure itself.

When all these minor comments are addressed, I think this paper can be considered for acceptance.

Reviewer #2: This study by Weixi Wu et al., mechanistically demonstrated the protective effect of GLP-1 RA against lipototoxity in diabetic renal injury. While this study may be of great interest to the scientific community, there are concerns to address.

* L26-31. It looks to me that the aim of this study is not well described. Are the authors investigated the mechanical protective role of GLP-1 RA or investigated the cholesterol accumulation and pyroptosis in glomerular endothelial cells (GEnCs)? If it is the latter, then it does not clearly match the title. Please clarify and make it consistent with the main result that drive the title of the manuscript.

* L51-53. Please improve the logical flow of the manuscript. I suggest the authors to write the second sentence first, then the first after: "Diabetic microvascular complications including DKD, leads to end stage renal disease [1]. In fact, globally, diabetic kidney disease (DKD) is a serious health and economic problem [2].

* L53-55. Please rewrite the following "The current approaches to preventing or treating DKD have limited effectiveness, such as controlling blood glucose levels, blood pressure, and urinary protein excretion" as: "The current approaches to prevent or treat DKD, which includes controlling blood glucose levels, blood pressure, and urinary protein excretion, have limited effectiveness. This limited effectiveness can be attributed to the lack of knowledge regarding the pathophysiological mechanism involved.

* L58-59. hard to understand sentence: not [... and triglyceride and cholesterol accumulated in the kidney of diabetes models is thought possibly contribute to DKD pathogenesis] BUT [... and accumulation of triglyceride and cholesterol in the kidney of diabetes models potentially contribute to DKD pathogenesis] or [... and accumulation of triglyceride and cholesterol in the kidney of diabetes models is thought to contribute to DKD pathogenesis].

* L69-70. Hard-to-understand sentence [To our knowledge, ATP-binding cassette transporter A1(ABCA1), which serve as transporters that promote cholesterol efflux from cell]. Please, rewrite it simply but accurately, or improve punctuation for a less confusing writing.

* "renoprotective or "reno-protective"? Please be consistent.

* L65. no need to repeat " interleukin-18" just right IL-18.

* It is suggested that the authors provide Ethics numbers or a copy of university ethical approval statement with institutional stamp.

* L92. Suggested to change "Cellular experiments" with "Cell culture experiments"

* L94-97. The cell cultures were divided into different growth conditions, including low glucose (LG, 5.5 mmol/L D-glucose ), high glucose (HG, 25.5 mmol/L D-glucose), high cholesterol (HC, 400 µg/ml watersoluble cholesterol) (Sigma, United States)], and HG combined HC (25.5 mmol/L+400 µg/ml).

* L198. [Values are shown as "mean + SD"]

* Captions in Figures only described what experiments have been performed, but not what each figure actually tells as information that answers the scientific question (the aim) of the study. Figures are meaningless/speechless. Figures, supported with captions should stand alone and provide enough information in order, such that we do not need to read the main text to get the main idea delivered by a figure.

* 205-207. Hard to understand sentence, hampering the result presentation.

* 207-209. The authors said that [The Oil red O staining and total cholesterol quantification showed that cholesterol accumulation "was not remarkable" in HG compared with LG, but increased under HC], but latter said that [these results showed that high glucose might exacerbate the accumulation of intracellular cholesterol under the load of cholesterol]. It looks contradictory to me. Rather, the results shows that the combination of HG and HC is required/associated with cholesterol accumulation, but not only HG alone. However, when we read the following data, it showed that HG alone or together with HC induce cholesterol accumulation and proinflammatory response, through inhibition of ABCA1. The authors should take care of the following sentence [The Oil red O staining and total cholesterol quantification showed that cholesterol accumulation "was not remarkable" in HG compared with LG, but increased under HC], which, I fell, is contradictory.

*L232. rewrite as follows "This heat map presented in Figure 3A displays the microarray data of aberrantly expressed circRNAs."

* I will stop the review at this stage, as and suggest the authors, to take the following together with what precede into accounts:

*** Overall: I would suggest the authors to improve the writeup by fixing the grammatical errors and syntax, and doublechecking the punctuations. This manuscript suffers from serious logical flow, from a sentence to another, and from two consecutive paragraph; this needs to be taken care of. I suggest the authors to seek for help from a scientific, with professional and proficient English.

The methods section needs to be written with clear details that can allow repeatability of the study. English syntax hampers understanding of the shared science. The authors should redo the data analysis. If contradictory results are found, need to stand as limitations of the study or provide better scientific explanations.

6. PLOS authors have the option to publish the peer review history of their article (what does this mean?). If published, this will include your full peer review and any attached files.

Reviewer #1: No

Reviewer #2: **Yes: **Arnaud John KOMBE KOMBE

---

## [Author Response · Author response to Decision Letter 0]

5 Jul 2024

Dear editor and reviewers, 

We would like to express our sincere gratitude to you for your efforts in handling our manuscript. We appreciate you give us the opportunity to revise our paper. We thank the editorial boards and the two reviewer’s points and suggestions to improve our work. Those comments are all valuable and very helpful for revising and improving our paper. 

We have studied comments carefully and have made correction which we hope meet with approval. Based on your comment and request, we attached revised manuscript with the correction sections red marked. We also replied the reviewers’ questions point-by-point as following below. The corrections in the paper and the responds to the reviewer’s comments are as following. 

Should you have any questions, please contact us without hesitate.

Reviewer #1:

These paper about “GLP-1 RA improves diabetic renal injury by alleviating Glomerular Endothelial Cells Pyrotosis via RXRα/circ8411/miR-23a-5p/ABCA1 pathway “suggesting that excessive hyperglycemia causes lipotoxic renal damage through the RXRα/circ8411/miR-23a-5p/ABCA1 43 pathway. Individuals suffering from hypercholesterolemia and diabetic kidney disease (DKD) may find relief by focusing on this pathway. The paper is well planned and written to put all the relevant data together. There are few minor corrections to be done. When all these minor comments are addressed, I think this paper can be considered for acceptance.

Response: We express our gratitude for your meticulous and constructive review, as well as your insightful comments. Your comments were highly insightful and has significantly enhanced the overall quality of our manuscript. According with your suggestions, we have made necessary revisions to the relevant section of the article. Subsequently, we present our point-by-point responses to each of your comments in the following pages.

1. The font size throughout the paper is not the same.

2. The spacings in the paper should be modified to be the same throughout the paper.

3. It would be better if you can reduce the thickness of the error bars in all the figures.

4. In Fig 7K immunostaining panel please mention blue and red in the figure itself.

Response: We appreciate the valuable advice provided and regret that the details of the article were not detailed enough. We have thoroughly reviewed and made revisions to this section. We've resized the body font to a uniform size, and doubled the line spacing as follows the manuscript body formatting guidelines.

Revised version in line 419-433 (Results): 

Figure 7. GLP-1RA alleviated kidney damage, cholesterol accumulation and pyroptosis in renal of ApoE-/-DM mice by increasing circ8411-ABCA1. (A) HE staining of the renal of mice (×400, bar=50μm). (B) Oil red O staining of the renal of mice (×400, bar=50μm). (C) Cholesterol quantification experiment to determine the cholesterol accumulation in renal of mice (**P < 0.01 vs. WT-NC, ***P < 0.001 vs. WT-NC, #P < 0.05 vs. ApoE-/-DM, ##P < 0.01 vs. ApoE-/-DM).(D,E) Caspase-1, GSDMD, IL-1β expression of renal of mice (*P < 0.05 vs. WT-NC, **P < 0.01 vs. WT-NC, ***P < 0.001 vs. WT-NC, #P < 0.05 vs. ApoE-/-DM, ##P < 0.01 vs. ApoE-/-DM).（F) Release of LDH of renal of mice (**P < 0.01 vs. WT-NC, ##P < 0.01 vs. ApoE-/-DM). (G) Circ8411 expression of renal of mice (*P < 0.05 vs. WT-NC, ##P < 0.01 vs. ApoE-/-). (H-J) RT-qPCR, Western Blot and immunofluorescence experiment (×200, bar=50μm) were conducted to determine ABCA1(red) expression of renal of mice. Nuclei were stained with DAPI (blue). (**P < 0.01 vs. WT-NC, ***P < 0.001 vs. WT-NC, ##P < 0.01 vs. ApoE-/-, ###P < 0.001 vs. ApoE-/-, &P < 0.05 vs. WT-NC).

Reviewer#2

This study by Weixi Wu et al., mechanistically demonstrated the protective effect of GLP-1 RA against lipototoxity in diabetic renal injury. While this study may be of great interest to the scientific community, there are concerns to address.

Response: We sincerely thank for your systematic guidance of my manuscript. Your comments were highly insightful, which greatly improved the content of this article as well as its framework and writing standards. According with your suggestions, we have amended the corresponding part of the article. In the following pages are our point-by-point responses to each of the comments.

1. L26-31. It looks to me that the aim of this study is not well described. Are the authors investigated the mechanical protective role of GLP-1 RA or investigated the cholesterol accumulation and pyroptosis in glomerular endothelial cells (GEnCs)? If it is the latter, then it does not clearly match the title. Please clarify and make it consistent with the main result that drive the title of the manuscript.

Response: We appreciate the valuable advice provided and express regret for the lack of clarity in the abstract. We have thoroughly reviewed and made revisions to this section.

Revised version in line 26-32(Methods): 

In this study, we investigated the cholesterol accumulation and pyroptosis in glomerular endothelial cells (GEnCs) and the reno-protective effects of GLP-1RAs using various techniques such as RT-qPCR, oil red O staining，Western blotting, and LDH activity assay. We also employed circRNA microarrays, bioinformatics analysis, gain and loss of function experiments, rescue experiments, and luciferase assay to identify the ceRNA mechanism involved. C57BL/6J, ApoE-/- mice were used for in vivo experiments.

2. L51-53. Please improve the logical flow of the manuscript. I suggest the authors to write the second sentence first, then the first after: "Diabetic microvascular complications including DKD, leads to end stage renal disease [1]. In fact, globally, diabetic kidney disease (DKD) is a serious health and economic problem [2].

Response: Thank you for bringing this to our attention. We sincerely appreciate your careful reading. We have diligently examined and refined the manuscript, encompassing the introduction through to the references.

Revised version in line 52-54(Introduction): 

Diabetic microvascular complications include diabetic kidney disease (DKD), which leads to end-stage renal disease [1]. Globally, DKD is a serious health and economic problem [2]. 

Revised version in line 606-613(References): 

1. de Boer IH, Khunti K, Sadusky T et al. (2022) Diabetes management in chronic kidney disease: a consensus report by the American Diabetes Association (ADA) and Kidney Disease: Improving Global Outcomes (KDIGO). Kidney Int 102: 974-989. https://doi.org/10.1016/j.kint.2022.08.012

2. Deng Y, Li N, Wu Y et al. (2021) Global, Regional, and National Burden of Diabetes-Related Chronic Kidney Disease From 1990 to 2019.Front Endocrinol (Lausanne) 12: 672350. https://doi.org/10.3389/fendo.2021.672350

3.L53-55. Please rewrite the following "The current approaches to preventing or treating DKD have limited effectiveness, such as controlling blood glucose levels, blood pressure, and urinary protein excretion" as: "The current approaches to prevent or treat DKD, which includes controlling blood glucose levels, blood pressure, and urinary protein excretion, have limited effectiveness. This limited effectiveness can be attributed to the lack of knowledge regarding the pathophysiological mechanism involved.

Response: We sincerely appreciate your careful reading. We think this is an excellent suggestion. We have re-edited the paragraph according to your suggestions.

Revised version in line 54-57(Introduction): 

The current approaches to prevent or treat DKD, which includes controlling blood glucose levels, blood pressure, and urinary protein excretion, have limited effectiveness. This limited effectiveness can be attributed to the lack of knowledge regarding the pathophysiological mechanism involved[3].

4.L58-59. hard to understand sentence: not [... and triglyceride and cholesterol accumulated in the kidney of diabetes models is thought possibly contribute to DKD pathogenesis] BUT [... and accumulation of triglyceride and cholesterol in the kidney of diabetes models potentially contribute to DKD pathogenesis] or [... and accumulation of triglyceride and cholesterol in the kidney of diabetes models is thought to contribute to DKD pathogenesis].

Response: Thank you for your valuable input. We have re-edited the paragraph according to your suggestions.

Revised version in line 58-60(Introduction): 

Studies have confirmed that abnormal lipid metabolism is strongly associated with DKD, and accumulation of triglyceride and cholesterol in the kidney of diabetes models potentially contribute to DKD pathogenesis[4-6].

5.L69-70. Hard-to-understand sentence [To our knowledge, ATP-binding cassette transporter A1(ABCA1), which serve as transporters that promote cholesterol efflux from cell]. Please, rewrite it simply but accurately, or improve punctuation for a less confusing writing.

Response: Thank you for your valuable input. We apologize for the poor language of our manuscript. We have now worked on both language and readability and have also involved native English speakers for language corrections. We really hope that the flow and language level have been substantially improved. We have re-edited the paragraph according to your suggestions.

Revised version in line 71-73(Introduction): 

The ATP-binding cassette transporter A1 (ABCA1) acts as a transporter that facilitates the removal of cholesterol from cells, by promoting cholesterol efflux.

6. "renoprotective or "reno-protective"? Please be consistent.

Response: We feel sorry for our carelessness. In our resubmitted manuscript, the typo is revised. Thanks for your correction.

Revised version in line 23-25(Background):

Furthermore, the mechanisms underlying any potential reno-protective effects of GLP-1 receptor agonists (GLP-1RAs) have not yet been established.

Revised version in line 26-29(Methods):

In this study, we investigated the cholesterol accumulation and pyroptosis in glomerular endothelial cells (GEnCs) and the reno-protective effects of GLP-1RAs using various techniques such as RT-qPCR, oil red O staining，Western blotting, and LDH activity assay.

Revised version in line 89-92(Introduction):

Taken together, we hypothesize that the GLP-1RA could exerts its reno-protective effects by regulating lipotoxicity-induced GEnCs pyroptosis via regulating ABCA1 in DKD and circRNAs may be involved in this procedure.

Revised version in line 554-556(Results):

In our study, we found that liraglutide and loxenatide has reno-protective and lipid-regulating effect in ApoE-/- DM mouse.

Revised version in line 578-581(Discussion)

More importantly, we first identify that liraglutide and loxenatide achieves reno-protection by reducing cholesterol accumulation and pyroptosis at least in part through RXRα/circ8411/miR-23a-5p/ABCA1 pathway under diabetic conditions.

7. L65. no need to repeat " interleukin-18" just right IL-18.

Response: Thank you for your valuable input. We have re-edited the paragraph according to your suggestions.

Revised version in line 65-68(Introduction):

Activating caspase-1 triggers the process of pyroptosis, inherently inducing inflammation and releasing pro-inflammatory factors, such as interleukin-1β (IL-1β), IL-18, lactate dehydrogenase (LDH) and the activation of a stronger inflammatory response[10].

8. It is suggested that the authors provide Ethics numbers or a copy of university ethical approval statement with institutional stamp.

Response: Thank you for your valuable input. We will upload a a copy of university ethical approval statement with institutional stamp.

9. L92. Suggested to change "Cellular experiments" with "Cell culture experiments"

Response: Thank you for your valuable input. We have re-edited the paragraph according to your suggestions.

Revised version in line 94(Materials and methods):

Cell culture experiments

10. L94-97. The cell cultures were divided into different growth conditions, including low glucose (LG, 5.5 mmol/L D-glucose ), high glucose (HG, 25.5 mmol/L D-glucose), high cholesterol (HC, 400 µg/ml watersoluble cholesterol) (Sigma, United States)], and HG combined HC (25.5 mmol/L+400 µg/ml).

Response: Thank you for your valuable input. Based on your comments, we have made the corrections and attached it below.

Revised version in line 96-99(Cell culture experiments):

The cell cultures were divided into different growth conditions, including low glucose (LG, 5.5 mmol/L D-glucose ), high glucose (HG, 25.5 mmol/L D-glucose), high cholesterol (HC, 400 µg/ml watersoluble cholesterol) (Sigma, United States)], and HG combined HC (25.5 mmol/L+400 µg/ml).

11. L198. [Values are shown as "mean + SD"]

Response: Thank you for your valuable input. We have re-edited the paragraph according to your suggestions.

Revised version in line 210(Statistical Analysis):

Values are shown as the "mean + SD".

12. Captions in Figures only described what experiments have been performed, but not what each figure actually tells as information that answers the scientific question (the aim) of the study. Figures are meaningless/speechless. Figures, supported with captions should stand alone and provide enough information in order, such that we do not need to read the main text to get the main idea delivered by a figure.

Response: Thank you for your valuable input. We've re-edited the captions.

Revised version in line 258-260;272-273(Results):

Figure 1. High glucose exacerbates intracellular cholesterol accumulation and pyroptosis under cholesterol load in GEnCs. (A) CCK-8 assay was used to evaluate the influence of water-soluble cholesterol on GEnCs viability (*P < 0.05 vs. Control; **P < 0.01 vs. Control,***P < 0.001 vs. Control). 25.5 mmol/L glucose and 400 mg/ml cholesterol were found to be the best intervention concentrations for HK-2 cells. (B) Oil red O staining of GEnCs treated with different concentrations of cholesterol (×400, bar=50μm). (C) Cholesterol quantification experiment was conducted to determine the content of intracellular cholesterol (*P < 0.05 vs. Control; **P < 0.01 vs. Control). (D) Expression of ABCA1 in GEnCs with different cholesterol concentrations (**P < 0.01 vs. Control,***P < 0.001 vs. Control). (E,F) Expression of ABCA1 after intervention with different glucose concentrations without cholesterol and under cholesterol load (*P < 0.05 vs. Control). (G) Oil red O staining of GEnCs treated with different concentrations of cholesterol and glucose (×400, bar=50μm). (H) Cholesterol quantification experiment was conducted to determine the content of intracellular cholesterol (*P < 0.05 vs. LG,**P < 0.01 vs. LG; ***P < 0.001vs. LG). (I) RT-qPCR analysis to determine ABCA1 mRNA expression (***P < 0.001 vs. LG; #P < 0.05 vs. HC). (J) Western Blot analysis to determine ABCA1 protein expression (***P < 0.001 vs. LG; #P < 0.05 vs. HC). ABCA1 levels decreased in the HG group and increased significantly in the HC group. (K) Immunofluorescence experiment was conducted to determine the expression of ABCA1 (×200, bar=100μm).

Revised version in line 280-281(Results):

Figure 2. HG exacerbates intracellular cholesterol accumulation and pyroptosis under cholesterol load. (A,B) RT-qPCR and Western Blot analysis to determine expressions of caspase-1, GSDMD, IL-1β in GEnCs under the condition of high glucose or/and high cholesterol (*P < 0.05 vs. LG,**P < 0.01 vs. LG; #P < 0.05 vs.HC, ##P < 0.01 vs.HC). Significant increase of pyroptosis markers were conserved in HG+HC than in HG or HC. (C) The release of LDH of GEnCs treated with different concentrations of cholesterol and glucose ((*P < 0.05 vs. LG,**P < 0.01 vs. LG; ##P < 0.01 vs.HC). (D,E)RT-qPCR and Western Blot analysis to determine ABCA1 expression of GEnCs treated with different concentrations of DIDS (*P < 0.05 vs. Control,***P < 0.001 vs. Control). (F) Oil red O staining of GEnCs treated with DIDS (×400, bar=50μm). (G) Cholesterol quantification experiment was conducted to determine the content of intracellular cholesterol (*P < 0.05 vs. Control). (H,I) RT-qPCR and Western Blot analysis to determine expressions of caspase-1, GSDMD, IL-1β in GEnCs treated with DIDS (*P < 0.05 vs. Control; **P < 0.01 vs. Control, ***P < 0.001 vs. Control). (J) Release of LDH of GEnCs treated with DIDS (****P < 0.001 vs. Control).

Revised version in line 316-317;323-324(Re

---

## [Decision Letter · Decision Letter 1]

2 Oct 2024

PONE-D-24-13952R1GLP-1 RA improves diabetic renal injury by alleviating Glomerular Endothelial Cells Pyrotosis via RXRα/circ8411/miR-23a-5p/ABCA1 pathwayPLOS ONE

Dear Dr. Yu,

Thank you for submitting your manuscript to PLOS ONE. After careful consideration, we feel that it has merit but does not fully meet PLOS ONE’s publication criteria as it currently stands. Therefore, we invite you to submit a revised version of the manuscript that addresses the points raised during the review process.

We look forward to receiving your revised manuscript.

Kind regards,

Miquel Vall-llosera Camps

Senior Staff Editor

PLOS ONE

Journal Requirements:

Reviewers' comments:

Reviewer's Responses to Questions

**Comments to the Author**

1. If the authors have adequately addressed your comments raised in a previous round of review and you feel that this manuscript is now acceptable for publication, you may indicate that here to bypass the “Comments to the Author” section, enter your conflict of interest statement in the “Confidential to Editor” section, and submit your "Accept" recommendation.

Reviewer #1: All comments have been addressed

Reviewer #2: All comments have been addressed

Reviewer #3: (No Response)

2. Is the manuscript technically sound, and do the data support the conclusions?

Reviewer #1: Yes

Reviewer #2: Yes

Reviewer #3: Yes

3. Has the statistical analysis been performed appropriately and rigorously? 

Reviewer #1: Yes

Reviewer #2: Yes

Reviewer #3: I Don't Know

4. Have the authors made all data underlying the findings in their manuscript fully available?

Reviewer #1: Yes

Reviewer #2: Yes

Reviewer #3: Yes

5. Is the manuscript presented in an intelligible fashion and written in standard English?

Reviewer #1: Yes

Reviewer #2: Yes

Reviewer #3: No

6. Review Comments to the Author

Reviewer #1: I think the author had addressed all the comments in a well manner and the paper can be accepted.

Congratulations to the author.

Reviewer #2: (No Response)

Reviewer #3: I appreciate your efforts in revising the manuscript, but several areas still require significant attention. Language and grammar issues persist throughout, and I strongly recommend professional editing to enhance clarity. Additionally, your responses to the reviewers' comments lack specificity. Consistent terminology (e.g reno-protective) and figure caption (too descriptive) are also necessary. A clearer point-by-point response will be beneficial in guiding your revisions (and do not need to cite the tools so many times)

7. PLOS authors have the option to publish the peer review history of their article (what does this mean?). If published, this will include your full peer review and any attached files.

Reviewer #1: No

Reviewer #2: **Yes: **Arnaud John KOMBE KOMBE

Reviewer #3: No

---

## [Author Response · Author response to Decision Letter 1]

16 Oct 2024

Dear editor and reviewers, 

We would like to express our sincere gratitude to you for your efforts in handling our manuscript. We appreciate you give us the opportunity to revise our paper. We thank the editorial boards and the reviewer’s points and suggestions to improve our work. Those comments are all valuable and very helpful for revising and improving our paper. 

We have studied comments carefully and have made correction which we hope meet with approval. Based on your comment and request, we attached revised manuscript with the correction sections red marked. The corrections in the paper and the responds to the reviewer’s comments are as following. 

Should you have any questions, please contact us without hesitate.

Reviewer #3: I appreciate your efforts in revising the manuscript, but several areas still require significant attention. Language and grammar issues persist throughout, and I strongly recommend professional editing to enhance clarity. Additionally, your responses to the reviewers' comments lack specificity. Consistent terminology (e.g reno-protective) and figure caption (too descriptive) are also necessary. A clearer point-by-point response will be beneficial in guiding your revisions (and do not need to cite the tools)

Response: We apologize for the presence of grammatical and typographical errors in our article, as they should have been avoided. We express our gratitude for the corrections provided by the expert. In order to refine the language and rectify any remaining errors, we have submitted them to a professional editing service. We attached revised manuscript with the correction sections red marked. Based on your suggestions, we have simplified the figure caption.

Revised version in line 26,29(Abstract) : 

 Lipotoxicity has been implicated in diabetic kidney disease (DKD). However, the role of high glucose levels in this process and the underlying mechanisms of renal protective effects of GLP-1 receptor agonists (GLP-1RAs) remain unclear.

To investigate cholesterol accumulation, pyroptosis in glomerular endothelial cells (GEnCs), and the renal protective effects of GLP-1RAs, we employed various techniques, including RT-qPCR, Oil Red O staining, Western blotting, lactate dehydrogenase (LDH) activity assays, circRNA microarrays, bioinformatics analysis, gain- and loss-of-function experiments, rescue experiments, and luciferase assays. Additionally, in vivo experiments were conducted using C57BL/6J and ApoE-/- mice.

Revised version in line 613(Discussion) : 

 In our study, we observed that liraglutide and loxenatide exhibited renal protective and lipid-regulating effects in ApoE-/-DM mice. These GLP-1RAs decreased cholesterol accumulation in ApoE-/-DM mice and GEnCs treated with high glucose and high cholesterol. 

Revised version in Results : 

Figure 1. High glucose exacerbates intracellular cholesterol accumulation and pyroptosis under cholesterol load in GEnCs. (A) CCK-8 assay was used to evaluate the influence of water-soluble cholesterol on GEnCs viability (*P < 0.05 vs. Control; **P < 0.01 vs. Control, ***P < 0.001 vs. Control). (B) Oil red O staining of GEnCs treated with different concentrations of cholesterol (×400, bar=50μm). (C) Cholesterol quantification experiment was conducted to determine the content of intracellular cholesterol (*P < 0.05 vs. Control; **P < 0.01 vs. Control). (D) Expression of ABCA1 in GEnCs with different cholesterol concentrations (**P < 0.01 vs. Control, ***P < 0.001 vs. Control). (E,F) Expression of ABCA1 after intervention with different glucose concentrations without cholesterol and under cholesterol load (*P < 0.05 vs. Control). (G) Oil red O staining of GEnCs treated with different concentrations of cholesterol and glucose (×400, bar=50μm). (H) Cholesterol quantification experiment was conducted to determine the content of intracellular cholesterol (*P < 0.05 vs. LG, **P < 0.01 vs. LG; ***P < 0.001vs. LG). (I) RT-qPCR analysis to determine ABCA1 mRNA expression (***P < 0.001 vs. LG; #P < 0.05 vs. HC). (J) Western Blot analysis to determine ABCA1 protein expression (***P < 0.001 vs. LG; #P < 0.05 vs. HC). (K) Immunofluorescence experiment was conducted to determine the expression of ABCA1 (×200, bar=100μm).

Figure 2. HG exacerbates intracellular cholesterol accumulation and pyroptosis under cholesterol load. (A,B) RT-qPCR and Western Blot analysis to determine expressions of caspase-1, GSDMD, IL-1β in GEnCs under the condition of high glucose or/and high cholesterol (*P < 0.05 vs. LG,**P < 0.01 vs. LG; #P < 0.05 vs. HC, ##P < 0.01 vs. HC). (C) The release of LDH of GEnCs treated with different concentrations of cholesterol and glucose ((*P < 0.05 vs. LG, **P < 0.01 vs. LG; ##P < 0.01 vs. HC). (D,E) RT-qPCR and Western Blot analysis to determine ABCA1 expression of GEnCs treated with different concentrations of DIDS (*P < 0.05 vs. Control,***P < 0.001 vs. Control). (F) Oil red O staining of GEnCs treated with DIDS (×400, bar=50μm). (G) Cholesterol quantification experiment was conducted to determine intracellular cholesterol content (*P < 0.05 vs. Control). (H,I) RT-qPCR and Western Blot analysis to determine expressions of caspase-1, GSDMD, IL-1β in GEnCs treated with DIDS (*P < 0.05 vs. Control; **P < 0.01 vs. Control, ***P < 0.001 vs. Control). (J) Release of LDH of GEnCs treated with DIDS (****P < 0.001 vs. Control).

Figure 3. Circ8411 is involved in the regulation of ABCA1. (A) The heatmap for the differential expression of circRNAs. (B) RT-qPCR analysis to determine the expression of four candidate lncRNAs (*P < 0.05 vs. HC; **P < 0.01 vs. HC). (C) Circ8411 expression of GEnCs treated with cholesterol and glucose (***P < 0.001 vs. LG, ###P < 0.001 vs. HC). (D) RT-qPCR to determine the knockdown efficiency of circ8411 (*P < 0.05 vs. si-NC). (E,F) Oil red O staining of GEnCs (×400, bar=20μm) and cholesterol quantification experiment to determine the cholesterol accumulation in cells transfected with siRNA. (G-I) RT-qPCR, Western Blot, and immunofluorescence experiment (×200, bar=50μm) were conducted to determine the expression of ABCA1 in cells transfected with siRNA (*P < 0.05 vs. NC-si; **P < 0.01 vs. NC-si). (J.K) RT-qPCR and Western Blot analysis to determine expressions of caspase-1, GSDMD, IL-1β in GEnCs treated with siRNA (*P < 0.05 vs. NC-si; **P < 0.01 vs. NC-si, ***P < 0.001 vs. NC-si). (L) Release of LDH of GEnCs treated with siRNA (*P < 0.05 vs. NC-si). (M) RT-qPCR analysis to determine the overexpression efficiency of circ8411 (***P < 0.001 vs. vector); (N,O) Oil red O staining of GEnCs (×400, bar=20μm) and cholesterol quantification experiment to determine the cholesterol accumulation in cells transfected with circ8411-pcDNA3.1 plasmid (**P < 0.01 vs. vector). (P,Q,S) RT-qPCR, Western Blot, and immunofluorescence experiment (×200, bar=50μm) were conducted to determine the expression of ABCA1 in cells transfected with the circ8411-pcDNA3.1 plasmid (*P < 0.05 vs. vector, **P < 0.01 vs. vector). (R,S) RT-qPCR and Western Blot analysis to determine the expression of caspase-1, GSDMD, and IL-1β in GEnCs treated with circ8411-pcDNA3.1 plasmid (*P < 0.05 vs. LG; **P < 0.01 vs. LG). (T) Release of LDH of GEnCs treated with circ8411-pcDNA3.1 plasmid (*P < 0.05 vs. vector).

Figure 4. The circ8411 regulates ABCA1 expression by regulating miR-23a-5p

(A) RT-qPCR analysis to determine miR-23a-5p in the GEnCs under conditions of high glucose and high cholesterol. (B,C) RT-qPCR analysis to determine miR-23a-5p expression with the knockdown and overexpression of circ8411. (**P < 0.05 vs. Control). (D,E) RT-qPCR analysis to determine the overexpression efficiency of miR-23a-5p and the expression of ABCA1 in cells transfected with the miR-23a-5p mimic (**P < 0.05 vs. Control). (F,G) Rescue experiments to determine the regulatory effect of circ8411/miR-23a-5p/ABCA1 in GEnCs (*P < 0.05 vs. mimic NC; #P < 0.05 vs. miR-23a-5p mimic with circ8411 vector). (H) Luciferase reporter assay on GEnCs co-transfected with circ8411-3’UTR-WT or circ8411-3’UTR-MUT and miR-23a-5p mimics or NC (**P < 0.01 vs. Circ8411-WT+NC group). 

Figure 5. High glucose affects circ8411 down-regulation through RXRs. (A) RT-qPCR analysis to determine the expression of RXRα, CEBP/α, Hif-1a in the GEnCs undering glucose and cholesterol (*P < 0.05 vs. Control, **P < 0.01 vs. Control). (B) Western Blot showed the expression of RXRα GEnCs under different intervention of glucose and cholesterol (*P < 0.05 vs. Control). (C) RXRα expression of GEnCs treated with different concentrations of UVI3003 (*P < 0.05 vs. Control). A marked reduction in RXRα was observed in cells treated with 10umol/L of UVI3003. (D) Circ8411 expression of GEnCs treated with UVI3003. (E) ABCA1 expression of GEnCs treated with UVI3003 (*P < 0.05 vs. Control). 

Figure 6. GLP-1RA attenuated cholesterol accumulation and pyroptosis in GEnCs by increasing RXRα-circ8411-miR-23a-5p-ABCA1. (A,B) CCK-8 assay was used to evaluate the influence of liraglutide and loxenatide on GEnC viability (***P < 0.001 vs. Control). (C,D) ABCA1 expression in the GEnCs treated with different concentrations of liraglutide and loxenatide (*P < 0.05 vs. Control; **P < 0.01 vs. Control). (E-H) Oil red O staining of GEnCs (×400, bar=50μm) and cholesterol quantification experiment to determine the cholesterol accumulation in cells treated with liraglutide and loxenatide (*P < 0.05 vs. Control). (I,J,L,M) Expressions of caspase-1, GSDMD, IL-1β in GEnCs treated with liraglutide and loxenatide (*P < 0.05 vs. Control; **P < 0.01 vs. Control). (K,N) Release of LDH of GEnCs treated with liraglutide and loxenatide (**P < 0.01 vs. Control). (O-T) Expressions of RXRα, circ8411, and miR-23a-5p of GEnCs treated with liraglutide and loxenatide (*P < 0.05 vs. Control, **P < 0.01 vs. Control). (U-V) Western Blot was performed to determine the ABCA1 expression in GEnCs treated with siRNAof circ8411 and GLP-1RA (*P < 0.05 vs. Control).

Figure 7. GLP-1RA alleviated kidney damage, cholesterol accumulation, and pyroptosis in the kidneys of ApoE-/-DM mice by increasing circ8411-ABCA1. (A) HE staining of the kidney tissues of mice (×400, bar=50μm). (B) Oil red O staining of the kidney tissues of mice (×400, bar=50μm). (C) Cholesterol quantification experiment to determine the cholesterol accumulation in the kidneys of mice (**P < 0.01 vs. WT-NC, ***P < 0.001 vs. WT-NC, #P < 0.05 vs. ApoE-/-DM, ##P < 0.01 vs. ApoE-/-DM).(D,E) Caspase-1, GSDMD, IL-1β expression in the kidneys of mice (*P < 0.05 vs. WT-NC, **P < 0.01 vs. WT-NC, ***P < 0.001 vs. WT-NC, #P < 0.05 vs. ApoE-/-DM, ##P < 0.01 vs. ApoE-/-DM). (F) Release of LDH in the kidneys of mice (**P < 0.01 vs. WT-NC, ##P < 0.01 vs. ApoE-/-DM). (G) Circ8411 expression of renal of mice (*P < 0.05 vs. WT-NC, ##P < 0.01 vs. ApoE-/-). (H-J) RT-qPCR, Western Blot, and immunofluorescence experiment (×200, bar=50μm) were conducted to determine ABCA1(red) expression in the kidneys of mice. Nuclei were stained with DAPI (blue). (**P < 0.01 vs. WT-NC, ***P < 0.001 vs. WT-NC, ##P < 0.01 vs. ApoE-/-, ###P < 0.001 vs. ApoE-/-, &P < 0.05 vs. WT-NC).

Once again, thank you very much for your constructive comments and suggestions which would help us both in English and in depth to improve the quality of the paper.

Yours sincerely,

Pei Yu

2024/10/16

---

## [Editor Report · Decision Letter 2]

29 Oct 2024

PONE-D-24-13952R2GLP-1 RA improves diabetic renal injury by alleviating Glomerular Endothelial Cells Pyrotosis via RXRα/circ8411/miR-23a-5p/ABCA1 pathwayPLOS ONE

Dear Dr. Yu,

Thank you for submitting your manuscript to PLOS ONE. After careful consideration, we feel that it has merit but does not fully meet PLOS ONE’s publication criteria as it currently stands. Therefore, we invite you to submit a revised version of the manuscript that addresses the points raised during the review process. While I appreciate the improvements made I have some concerns regarding its readiness for publication. The language still presents challenges, with several grammatical errors that affect clarity. Some sentences in the manuscript (in particular, in the introduction) are quite short and lead to difficulty in following the overall narrative. I recommend rephrasing them for better flow and coherence. In addition there are also ongoing inconsistencies in terminology and figure captions, which complicate the evaluation of the findings.

I recommend a thorough revision to address these issues before considering acceptance, as the article has interesting potential. For this, authors may seek independent editorial help before submitting the revision. These services can be found on the web using search terms like “scientific editing service” or “manuscript editing service.”

We look forward to receiving your revised manuscript.

Kind regards,

Joel Montané, PhD

Academic Editor

PLOS ONE
---

## [Author Response · Author response to Decision Letter 2]

11 Nov 2024

Dear editor and reviewers, 

We would like to express our sincere gratitude to you for your efforts in handling our manuscript. We appreciate you give us the opportunity to revise our paper. We thank the editorial boards and the reviewer’s points and suggestions to improve our work. Those comments are all valuable and very helpful for revising and improving our paper. 

We have studied comments carefully and have made correction which we hope meet with approval. Based on your comment and request, we attached revised manuscript with the correction sections red marked. The corrections in the paper and the responds to the reviewer’s comments are as following. 

Should you have any questions, please contact us without hesitate.

“While I appreciate the improvements made I have some concerns regarding its readiness for publication. The language still presents challenges, with several grammatical errors that affect clarity. Some sentences in the manuscript (in particular, in the introduction) are quite short and lead to difficulty in following the overall narrative. I recommend rephrasing them for better flow and coherence. In addition there are also ongoing inconsistencies in terminology and figure captions, which complicate the evaluation of the findings.

I recommend a thorough revision to address these issues before considering acceptance, as the article has interesting potential. ”

Response: We apologize for the presence of grammatical and typographical errors in our article, as they should have been avoided. In order to refine the language and rectify any remaining errors, we have submitted them to a professional editing service. We attached revised manuscript with the correction sections red marked. Based on your suggestions, we have thoroughly examine and revise the figure captions and the section of introduction.Additionally, we have re-edited the figures with the objective of enhancing clarity and comprehensibility.

Revised version in line 26,29,30,36,38,42,49(Abstract) : 

Lipotoxicity has been implicated in diabetic kidney disease (DKD). However, the role of high glucose levels in DKD and the underlying renal protective mechanisms of GLP-1 receptor agonists (GLP-1RAs) remain unclear.

To investigate cholesterol accumulation, pyroptosis in glomerular endothelial cells (GEnCs), and the renal protective mechanisms of GLP-1RAs, we used various techniques, including RT-qPCR, Oil Red O staining, Western blotting, lactate dehydrogenase (LDH) activity assays, circRNA microarrays, bioinformatics analysis, gain and loss-of-function experiments, rescue experiments, and luciferase assays. 

GEnCs exposed to high glucose exhibited reduced cholesterol efflux, which was accompanied by downregulation of ATP-binding cassette transporter A1 (ABCA1) expression, cholesterol accumulation, and pyroptosis. Circ8411 was identified as a regulator of ABCA1, inhibiting miR-23a-5p through its binding to the 3'UTR. Additionally, higher glucose levels decreased circ8411 expression by inhibiting RXRα. GLP-1RAs effectively reduced cholesterol accumulation and cell pyroptosis by targeting the RXRα/circ8411/miR-23a-5p/ABCA1 pathway. In diabetic ApoE-/- mice, renal structure and function were impaired, with resulted in increased cholesterol accumulation and pyroptosis; however, GLP-1RAs treatment reversed these detrimental changes.

Moreover, GLP-1RAs may provide reno-protective effects by activating this pathway.

Revised version in line (Introduction) : 

Diabetic kidney disease (DKD) is a leading cause of end-stage renal disease (ESRD) [1],which poses a significant health and economic burden [2]. Current strategies to prevent or treat DKD have limited efficacy, such as the management of blood glucose, blood pressure, and urinary protein excretion. This limitation can be attributed to an insufficient understanding of the underlying pathophysiological mechanisms [3]. 

Numerous studies have established a strong association between abnormal lipid metabolism and DKD. The accumulation of triglycerides(TG) and cholesterol in the kidneys has been implicated in the pathogenesis of DKD [4-6]. Lipotoxicity, which is characterized by the dysregulation of intracellular homeostasis due to lipid accumulation, leads to metabolic, inflammatory, and oxidative stress [7]. Hyperlipidemia, oxidized low-density lipoprotein (ox-LDL), and cholesterol crystals have been shown to activate caspase-1 in endothelial cells, thereby triggering pyroptosis [8, 9]. Pyroptosis is an inflammatory process that is characterised by the release of pro-inflammatory factors, including interleukin-1β (IL-1β), IL-18, and lactate dehydrogenase (LDH). These factors can contribute to a more pronounced inflammatory response[10]. However, the specific mechanisms underlying pyroptosis in the context of DKD development remain unclear.

There is an increasing consensus that glomerular endothelial cells (GEnCs) play a critical role in the development and progression of DKD [11-13]. The ATP-binding cassette transporter A1 (ABCA1) is a key regulator of cholesterol efflux from cells, promoting the removal of cholesterol [14-16]. Previous researches suggest that ABCA1 expression in GEnCs may help to prevent endothelial dysfunction [5, 17].

ABCA1 is a plasma membrane protein that serves as a key regulator of cholesterol efflux from cells[18, 19]. In vitro studies have demonstrated that stimulation of high glucose can downregulate ABCA1 expression in renal cells, and a similar decrease in ABCA1 has been observed in diabetic and nephrotic mouse models [20, 21]. Furthermore, our previous research indicated that the dysregulation of glucose and cholesterol metabolism can impair ABCA1 function [22]. Collectively, these findings suggest that dysregulation of ABCA1 may contribute to cholesterol accumulation in the kidney. 

Noncoding RNAs (ncRNAs) are a class of RNAs that do not encode proteins. A specific subset of ncRNAs are known as circular RNAs (circRNAs). CircRNAs are characterized by a covalently closed loop formed through a back-splicing process [23]. CircRNAs can act as microRNA (miRNA) sponges or interact with RNA-binding proteins, thereby regulating gene expression [24]. They have been identified as key regulators in various diseases [25]. However, the potential role of circRNAs in the progression of DKD remains unexplored. The glucose-like peptide-1 receptor (GLP-1R) and its agonists (GLP-1RAs) have shown protective effects against oxidative stress and macrophage activation in DKD [26]. Specifically, Exendin-4, a GLP-1RA, has been shown to ameliorate lipotoxicity-induced injury in GEnCs by enhancing ABCA1-mediated cholesterol efflux [27, 28]. Based on these findings, we hypothesize that GLP-1RAs may exert their renal protective effects in DKD by regulating lipotoxicity-induced pyroptosis in GEnCs through the modulation of ABCA1. Notably, circRNAs may play a significant role in this regulatory process.

Revised version in Results(238-254) : 

CCK8 assay and Oil Red O staining were employed to determine the optimal concentration and time for cholesterol. Based on the results, 400 μg/ml cholesterol and 24 hours were identified as optimal intervention concentration and time for GEnCs(Fig 1A-B). The treatment of GEnCs with varying concentrations of glucose and cholesterol demonstrated a gradual elevation in ABCA1 mRNA expression with increasing cholesterol levels(Fig 1D), and the reduction in ABCA1 mRNA was most pronounced when cells were treated with 25.5mmo/L glucose(Fig 1E-F). Accordingly, 400 µM cholesterol and 25 µM glucose were selected as the intervention concentrations for the subsequent cell experiments. When compared to high glucose or high cholesterol individually, intracellular cholesterol accumulation was significantly elevated in cells treated with a combination of high glucose and high cholesterol (Fig 1G, H). Furthermore, the results demonstrated that ABCA1 levels were downregulated in the HG group compared to LG and significantly upregulated in the HC group. ABCA1 expression was significantly downregulated in the HG and HC groups compared to HC (Fig 1I-K). In summary, these findings suggest that high glucose may exacerbate intracellular cholesterol accumulation under conditions of cholesterol overload, the process may be mediated by ABCA1.

Revised version in Results(419-421) : 

 Among these, CEBP/α and RXRα exhibited significantly decreased expression in GEnCs cultured in HG+HC group compared to those cultured in HC group (Fig 5A). RXRα was selected for further investigation in the subsequent study. To suppress RXRα expression effectively in GEnCs, we employed different concentrations of UVI3003. 

Revised version in Captions：

Figure 1. High glucose exacerbates intracellular cholesterol accumulation under cholesterol load in GEnCs. (A) CCK-8 assay was used to evaluate the influence of water-soluble cholesterol on GEnCs viability (*P < 0.05 vs. Cho-0μg/ml group; **P < 0.01 vs. Cho-0μg/ml group, ***P < 0.001 vs. Cho-0μg/ml group). (B)Representative images of Oil red O staining of GEnCs treated with different concentrations of cholesterol and the quantitative assessment(×400, bar=50μm; ***P < 0.001 vs. Cho-0μg/ml group). (C) Cholesterol quantification experiment was conducted to determine the content of intracellular cholesterol (*P < 0.05 vs. Cho-0μg/ml group; **P < 0.01 vs. Cho-0μg/ml group). (D)The mRNA expression of ABCA1 in GEnCs was detected by RT-qPCR with different cholesterol concentrations (**P < 0.01 vs. Cho-0 μg/ml group, ***P < 0.001 vs. Cho-0μg/ml group). (E) The mRNA expression of ABCA1 in GEnCs was detected by RT-qPCR after intervention with different glucose concentrations without cholesterol(*P < 0.05 vs. Glu-5.5mmol/L group). (F) The mRNA expression of ABCA1 in GEnCs was detected by RT-qPCR after intervention with different glucose concentrations under cholesterol load (#P < 0.05 vs. Cho-400μg/ml+Glu-5.5mmol/L group). (G) Representative images of Oil red O staining of GEnCs and the quantitative assessment(×400, bar=50μm; ##P < 0.01 vs. HC, ***P < 0.001 vs. LG group). (H) Cholesterol quantification experiment was conducted to determine the content of intracellular cholesterol (*P < 0.05 vs. LG group, **P < 0.01 vs. LG group; ***P < 0.001vs. LG group). (I) RT-qPCR analysis to determine ABCA1 mRNA expression (***P < 0.001 vs. LG group; #P < 0.05 vs. HC group). (J) Western Blot analysis to determine ABCA1 protein expression (***P < 0.001 vs. LG group; #P < 0.05 vs. HC group). (K) Representative images of Immunofluorescence staining of ABCA1(red) in GEnCs(×200, bar=100μm). Nucleus were stained with DAPI (blue).

Figure 2. HG exacerbates GEnCs pyroptosis through ABCA1 under cholesterol load. (A,B) RT-qPCR and Western Blot analysis to determine expressions of caspase-1, GSDMD, IL-1β in GEnCs under the condition of high glucose or/and high cholesterol (*P < 0.05 vs. LG group,**P < 0.01 vs. LG group; #P < 0.05 vs. HC group, ##P < 0.01 vs. HC group). (C) The release of LDH of GEnCs treated with different conditions(*P < 0.05 vs. LG group, **P < 0.01 vs. LG group; ##P < 0.01 vs. HC group). (D,E) RT-qPCR and Western Blot analysis to determine ABCA1 expression of GEnCs treated with different concentrations of DIDS (*P < 0.05 vs. HG+HC+DIDS-0μmmol/L group,***P < 0.001 vs. HG+HC+DIDS-0μmmol/L group). (F) Representative images of Oil red O staining and analysis of GEnCs treated with 400 μmol/L DIDS (×400, bar=50μm；**P < 0.01 vs. HG+HC group). (G) Cholesterol quantification experiment was conducted to determine intracellular cholesterol content (*P < 0.05 vs. HG+HC group). (H,I) RT-qPCR and Western Blot analysis to determine expressions of caspase-1, GSDMD, IL-1β in GEnCs treated with 400 μmol/L DIDS (*P < 0.05 vs. HG+HC group; **P < 0.01 vs. HG+HC group, ***P < 0.001 vs. HG+HC group). (J) Release of LDH of GEnCs treated with 400 μmol/L DIDS (***P < 0.001 vs. HG+HC group).

Figure 3. Circ8411 is involved in the regulation of ABCA1. Figure 3. Circ8411 is involved in the regulation of ABCA1. (A) The heatmap for the differential expression of circRNAs. (B) RT-qPCR analysis to determine the expression of four candidate lncRNAs (*P < 0.05 vs. HC group; **P < 0.01 vs. HC group). (C) Circ8411 expression of GEnCs treated with cholesterol and glucose (***P < 0.001 vs. LG group, ###P < 0.001 vs. HC group). (D) RT-qPCR to determine the knockdown efficiency of circ8411 (*P < 0.05 vs. NC-si group). (E) Representative images of Oil red O staining(×400, bar=20μm) and analysis in GEnCs after knockedown circ8411(***P < 0.001 vs. HG+HC+si-NC group). (F) Cholesterol quantification experiment to determine the cholesterol accumulation in GEnCs transfected with siRNA (*P < 0.05 vs. HG+HC+si-NC group). (G-I) RT-qPCR, Western Blot, and immunofluorescence experiment (×200, bar=50μm) were conducted to determine the expression of ABCA1 in cells transfected with siRNA (*P < 0.05 vs. NC-si group; **P < 0.01 vs. NC-si group). (J) RT-qPCR analysis to determine expressions of caspase-1, GSDMD, IL-1β in GEnCs treated with siRNA (*P < 0.05 vs. NC-si group; **P < 0.01 vs. NC-si group, ***P < 0.001 vs. NC-si group). (K)Representative Western blot images and quantitative data of caspase-1, GSDMD, IL-1β in GEnCs treated with siRNA (*P < 0.05 vs. HG+HC+si-NC group; **P < 0.01 vs. HG+HC+si-NC group, ***P < 0.001 vs. HG+HC+si-NC group). (L) Release of LDH in GEnCs treated with siRNA (*P < 0.05 vs. HG+HC+si-NC group). (M) RT-qPCR analysis to determine the overexpression efficiency of circ8411 (***P < 0.001 vs. cir8411-vector group). (N) Representative images of Oil red O staining(×400, bar=20μm) and analysis in GEnCs transfected with circ8411-pcDNA3.1 plasmid(**P < 0.01 vs. HG+HC+cir8411-vector group). (O) Cholesterol quantification experiment to determine the cholesterol accumulation in GEnCs transfected with circ8411-pcDNA3.1 plasmid (**P < 0.01 vs. HG+HC+cir8411-vector group). (P) RT-qPCR were conducted to determine the expression of ABCA1 mRNA in cells transfected with the circ8411-pcDNA3.1 plasmid (**P < 0.01 vs. HG+HC+cir8411-vector group). (Q)Representative images of the immunofluorescence staining of ABCA1(red) in cells transfected with the circ8411-pcDNA3.1 plasmid (×200, bar=50μm). Nucleus were stained with DAPI (blue). (R) RT-qPCR analysis to determine the expression of caspase-1, GSDMD, and IL-1β in GEnCs treated with circ8411-pcDNA3.1 plasmid (*P < 0.05 vs. cir8411-vector group, **P < 0.01 vs. cir8411-vector group). (S) Representative Western blot images and quantitative data of caspase-1, ABCA1, GSDMD, and IL-1β in GEnCs treated with circ8411-pcDNA3.1 plasmid (*P < 0.05 vs. HG+HC+cir8411-vector group; **P < 0.01 vs. HG+HC+cir8411-vector group). (T) Release of LDH in GEnCs treated with circ8411-pcDNA3.1 plasmid (*P < 0.05 vs. HG+HC+cir8411-vector group).

Figure 4. The circ8411 regulates ABCA1 expression by regulating miR-23a-5p

(A)RT-qPCR analysis to determine miR-23a-5p in the GEnCs under different conditions (**P < 0.01 vs. LG group；###P < 0.001 vs. HG group). (B) RT-qPCR analysis to determine miR-23a-5p expression with the knockdown of circ8411(**P < 0.01 vs. LG+si-NC group, ##P < 0.01 vs. HG+HC+si-NC group). (C) RT-qPCR analysis to determine miR-23a-5p expression with the overexpression of circ8411 (**P < 0.01 vs. LG+circ8411-vector group, ##P < 0.01 vs. HG+HC+circ8411-vector group). (D) RT-qPCR analysis to determine the overexpression efficiency of miR-23a-5p in cells treated with the miR-23a-5p mimic (***P < 0.001 vs. miR-23a-5p mimic-NC group). (E) RT-qPCR analysis to determine the expression of ABCA1 in cells treated with the miR-23a-5p mimic (**P < 0.01 vs. HG+HC+mimic-NC group). (F) Rescue experiments to determine the regulatory effect of circ8411/miR-23a-5p/ABCA1 in GEnCs (*P < 0.05 vs. HG+HC+mimic-NC group, #P < 0.05 vs. HG+HC+mimic-miR-23a-5p+circ8411 vector group). (G) Western blot analysis to determine expression of ABCA1 in rescue experiments(*P < 0.05 vs. HG+HC+mimic-NC group; #P < 0.05 vs. HG+HC+mimic-miR-23a-5p+circ8411 vector group). (H) Luciferase reporter assay on GEnCs co-transfected with circ8411-3’UTR-WT or circ8411-3’UTR-MUT and miR-23a-5p mimic or NC (**P < 0.01 vs. mimic-NC group). 

Figure 5. High glucose affects circ8411 down-regulation through RXRα. 

(A) RT-qPCR analysis to determine the expression of RXRα, CEBP/α

---

## [Editor Report · Decision Letter 3]

14 Nov 2024

GLP-1 RA improves diabetic renal injury by alleviating Glomerular Endothelial Cells Pyrotosis via RXRα/circ8411/miR-23a-5p/ABCA1 pathway

PONE-D-24-13952R3

Dear Dr. Yu,

We’re pleased to inform you that your manuscript has been judged scientifically suitable for publication and will be formally accepted for publication once it meets all outstanding technical requirements.

Kind regards,

Joel Montané, PhD

Academic Editor

PLOS ONE

Additional Editor Comments (optional):

The authors have addressed all comments, and the manuscript is now ready for publication.
---

## [Editor Report · Acceptance letter]

19 Nov 2024

PONE-D-24-13952R3 

PLOS ONE

Dear Dr. Yu, 

I'm pleased to inform you that your manuscript has been deemed suitable for publication in PLOS ONE. Congratulations! Your manuscript is now being handed over to our production team.

Kind regards, 

on behalf of

Dr. Joel Montané 

Academic Editor

PLOS ONE